# Binding kinetics drive G protein subtype selectivity at the β₁-adrenergic receptor

Andrew J. Y. Jones[1,3], Thomas H. Harman[1,3], Matthew Harris[2], Oliver E. Lewis ®[1], Graham Ladds ®[2] & Daniel Nietlispach ®[1] ✉

G protein-coupled receptors (GPCRs) bind to different G protein α-subtypes with varying degrees of selectivity. The mechanism by which GPCRs achieve this selectivity is still unclear. Using ¹³C methyl methionine and ¹⁹F NMR, we investigate the agonist-bound active state of β₁AR and its ternary complexes with different G proteins in solution. We find the receptor in the ternary complexes adopts very similar conformations. In contrast, the full agonist-bound receptor active state assumes a conformation differing from previously characterised activation intermediates or from β₁AR in ternary complexes. Assessing the kinetics of binding for the agonist-bound receptor with different G proteins, we find the increased affinity of β₁AR for Gs results from its much faster association with the receptor. Consequently, we suggest a kinetic-driven selectivity gate between canonical and secondary coupling which arises from differential favourability of G protein binding to the agonist-bound receptor active state.

G protein-coupled receptors (GPCRs) represent the largest family of membrane proteins in humans with over 800 members[1]. As cell surface receptors, GPCRs transduce a wide range of stimuli across membranes which triggers diverse downstream signalling cascades affecting many physiological processes[2]. Binding of agonists into the orthosteric ligand-binding pocket results in large conformational rearrangements of the receptor transmembrane domain, exposing a cytoplasmic pocket into which effectors bind[3].

GPCRs couple to several different intracellular effectors including heterotrimeric G proteins which interact with receptors primarily through their α-subunits. The 16 human Gα subtypes are grouped into four families, Gs, Gi/o, Gq/11, and G12/13[4]. Despite co-expression of multiple G proteins in cellular environments, many GPCRs couple selectively to a single G protein α-subtype with reduced secondary coupling to other subtypes[5,6]. In contrast, other GPCRs show promiscuous coupling to multiple G proteins[7,8]. As each G protein family activates distinct signalling cascades[2], delineating G protein selectivity mechanisms is essential to the fundamental understanding of cellular signalling and developing pathway specific GPCR therapeutics.

Spectroscopic methods, including solution nuclear magnetic resonance (NMR) spectroscopy, have established GPCRs as highly plastic entities capable of sampling an equilibrium of conformations[9–16]. Agonist binding influences the equilibrium populations of states in the GPCR activation pathway[17–23]. The GPCR active state, the conformation most proficient to engage with canonical binding partners, is inherently unstable and dynamic, preventing direct structural resolution. Consequently, the conformation observed in ternary complex structures has been assumed as a suitable approximation of the active state. However, recent NMR studies demonstrate structural differences between active and ternary complex states[24,25], highlighting the necessity for characterising receptors in solution. A rapidly growing number of GPCR structures in ternary complex with G proteins from different families have been solved by cryo-electron microscopy (cryo-EM)[26]. Comparisons reveal variation in G protein orientation, globally and in the critical C-terminal α5 helix[26], whilst receptor conformation differences are exemplified in intracellular loop 2 (IL2)[27–29], the transmembrane helix (TM) 5 and TM6 lengths[30], and the TM7/H8 hinge (IL4) region[27]. The magnitude of TM6 displacement has also been implicated in Gs/Gi selectivity[31–34].

---

[1]Department of Biochemistry, University of Cambridge, 80 Tennis Court Road, Cambridge CB2 1GA, UK. [2]Department of Pharmacology, University of Cambridge, Tennis Court Road, Cambridge CB2 1PD, UK. [3]These authors contributed equally: Andrew J. Y. Jones, Thomas H. Harman. ✉ e-mail: dn206@cam.ac.uk

However, comparisons of the same receptor bound to different G proteins show very minor differences in TM6 displacement[27,35]. Therefore, solution NMR studies of receptors in an active state coupling to different G proteins are essential to complement static structural studies in illuminating G protein selectivity but remain sparse[29].

The class A GPCR $\beta_1$-adrenergic receptor ($\beta_1$AR), the predominant $\beta$AR subtype in cardiac tissue, canonically couples to $G_s$ with secondary coupling to $G_i$[36–38]. In our previous work, we examined agonist dependent changes to receptor conformation and coupling to active state-stabilising nanobody Nb6B9 using a minimally thermostabilised construct $\beta_1$AR-Met$\Delta$5 (here $\beta_1$AR-W)[19,20]. We demonstrated ligand efficacy-dependent changes to equilibria between inactive and activation-intermediate (pre-active) receptor states but observed significant differences between agonist-bound and nanobody-bound ternary complex spectra. We hypothesised the specific set of thermostabilising mutations used significantly attenuated population of the active state, favouring the inactive and pre-active receptor states. Therefore, we further reverted the TM4-TM3-TM5 interface-stabilising E130W[3.41] mutation[39] (superscripts refer to Ballesteros-Weinstein numbering[40]) to wild type E130[3.41] (here, $\beta_1$AR-E) to restore accessibility to the agonist-bound active state.

In this study, we examine $\beta_1$AR activation and G protein selectivity in solution by characterising the structural features of the receptor agonist-bound active state and in complex with primary and non-canonical G$\alpha$ analogues using NMR spectroscopy. Our study shows the active state to be similar to the conformation found in ternary structures[41] but requires additional conformational rearrangements to reach the solution $G_s$ ternary complex. We further find that whilst subtle conformational differences between ternary complexes with different G proteins are indicative of receptor adaptability, corroboratory in vitro binding assays show rates of complex formation with $G_s$ are significantly more rapid. Consequently, we suggest a kinetic-driven selectivity gate between canonical and secondary coupling which arises from differential favourability of G protein binding to the receptor active state.

## Results

### $\beta_1$AR NMR constructs are competent for coupling to G proteins
In this work we reversed the E130W[3.41] mutation[39] in the $\beta_1$AR-W construct[20], producing the less thermostabilised construct $\beta_1$AR-E (Supplementary Fig. 1), to facilitate examination of the receptor active state and to evaluate ternary complex formation with G proteins by NMR spectroscopy. We assessed the functional integrity and pharmacological profiles of the $\beta_1$AR-E and thermostabilised $\beta_1$AR-W receptor constructs in cellular and biophysical assays examining ligand binding and agonist-dependent G protein coupling, validating the suitability of $\beta_1$AR-E for use in studying receptor activation. Studies examining $\beta_1$AR activation commonly use the therapeutic full agonist isoprenaline in place of the native ligand adrenaline[17,41–43]. Consequently, we measured isoprenaline binding affinities of the $\beta_1$AR constructs in HEK293T cells expressing receptors N-terminally tagged with Nluc[44] (Supplementary Fig. 2a). NanoBRET signal loss by competition of $\beta_1$AR antagonist (S)-propranolol-red[45] ($K_D$ - $\beta_1$AR-W: 2.81 ± 0.56 nM; $\beta_1$AR-E: 3.09 ± 0.30 nM, Supplementary Fig. 2b) with isoprenaline at varying concentrations demonstrated low micromolar $K_i$ binding with no significant difference ($p = 0.65$) in affinities between $\beta_1$AR-W and $\beta_1$AR-E (Fig. 1a), indicating ligand binding was little affected by this thermostabilising mutation.

We next assessed G protein coupling using the TRUPATH BRET2 biosensor assay[46] (Supplementary Fig. 2c) expressed in HEK293T cells, allowing examination of signalling via the canonical coupling partner $G_s$. Comparison of isoprenaline-stimulated concentration response curves showed that both $\beta_1$AR receptor constructs could activate $G_s$ (Fig. 1b). As expected, for a thermostabilised construct that favours the inactive states, the $\beta_1$AR-W construct displayed reduced ($p = 0.0098$) isoprenaline potency ($\beta_1$AR-W (pEC$_{50}$: 7.23 ± 0.06) compared to the less thermostablised construct $\beta_1$AR-E (pEC$_{50}$: 7.77 ± 0.10). Therefore, the TRUPATH and ligand binding assays confirmed that reversal of the thermostabilising mutation increased receptor activity without impacting ligand affinity. Also, although less efficacious, the thermostabilised $\beta_1$AR-W construct still retained functional G protein coupling.

We further characterised ternary complex formation in vitro using a bio-layer interferometry (BLI) assay. This technique facilitated determination of affinity ($K_D$) and kinetics of G protein binding ($k_{on}$, $k_{off}$) to $\beta_1$AR, under similar solution conditions used for NMR spectroscopy. Receptor purified in detergent was immobilised to streptavidin-coated biosensors via an N-terminal biotinylated avi-tag on the receptor (Fig. 1c, see Supplementary Fig. 3 for controls) to ensure the receptor's cytoplasmic binding surface remained accessible to bulky binding partners. In anticipation of NMR experiments, we examined ternary complex formation of isoprenaline-bound receptor with the engineered mini-$G_s$ protein[47,48] (Supplementary Fig. 2d). The lower molecular weight of mini-G proteins due to removal of the $\alpha$-helical domain makes them an attractive means to study ternary G protein complexes in solution by NMR, resulting in smaller signal linewidths and improved sensitivity. Binding of mini-$G_s$ to both $\beta_1$AR-E and $\beta_1$AR-W was observed (Fig. 1d) with the binding curves agreeing with a 1:1 interaction stoichiometry. Furthermore, it was observed that the $\beta_1$AR-E construct demonstrated higher mini-$G_s$ affinity (consistent with the results of the TRUPATH assay) primarily due to increased $k_{on}$ rate relative to $\beta_1$AR-W, confirming the increase in receptor activity following reversal of the thermostabilising mutation. Dose-response experiments confirmed the lower $K_D$ of $\beta_1$AR-E (Supplementary Fig. 4, Supplementary Table 1).

Minimal binding was observed in the absence of ligand (Supplementary Fig. 2e), indicating functional complex formation was agonist specific and basal receptor activity was low, consistent with the literature[49]. As anticipated, the application of apyrase (Supplementary Fig. 2f) had little effect upon mini-$G_s$ binding, since the mini-G proteins contained a mutation that made them insensitive to nucleotide binding when coupled to GPCRs[47]. Agonist-dependent binding to both constructs was also shown for the active state-stabilising nanobody, Nb6B9. Again, the more active construct $\beta_1$AR-E showed increased binding affinity compared to $\beta_1$AR-W (Supplementary Fig. 2g, Supplementary Table 2). Together, these data illustrated the functionality of receptor constructs both in vivo and purified in detergent and confirmed that the reversal of E130W[3.41] to wild type E130[3.41] resulted in a receptor with greater capacity for activation.

### Characterisation of the $\beta_1$AR-E solution active state by NMR
We hypothesised that the more active $\beta_1$AR-E construct would access the active state with a much higher occupancy than in our previous work with $\beta_1$AR-W[19,20]. Therefore, we used the $\beta_1$AR-E construct to characterise the conformational properties of the active state in solution by NMR spectroscopy. We labelled $\beta_1$AR-E with $^{13}$C methyl methionine, facilitating examination of the NMR response to full agonist on the natively occurring methionine positions M153[34.57] (IL2), M178[4.62] (EL2), M223[5.54] (TM5), M283[6.28] (TM6), and M296[6.41] (TM6) (Fig. 2a) assigned in previous work using $\beta_1$AR-W[20]. We assessed conformational changes of the receptor upon isoprenaline addition through 2D $^1$H-$^{13}$C HMQC methionine NMR shift correlation experiments and used peak changes in both dimensions compared to the apo state spectrum to inspect differences in conformations and dynamics (Supplementary Fig. 5). Overlaying $\beta_1$AR-E apo and isoprenaline-bound spectra revealed significant differences in peak positions that were indicative of conformational changes across the whole receptor upon agonist binding (Fig. 2b). The affected residues changed their peak positions in directions indicative of overall receptor activation as

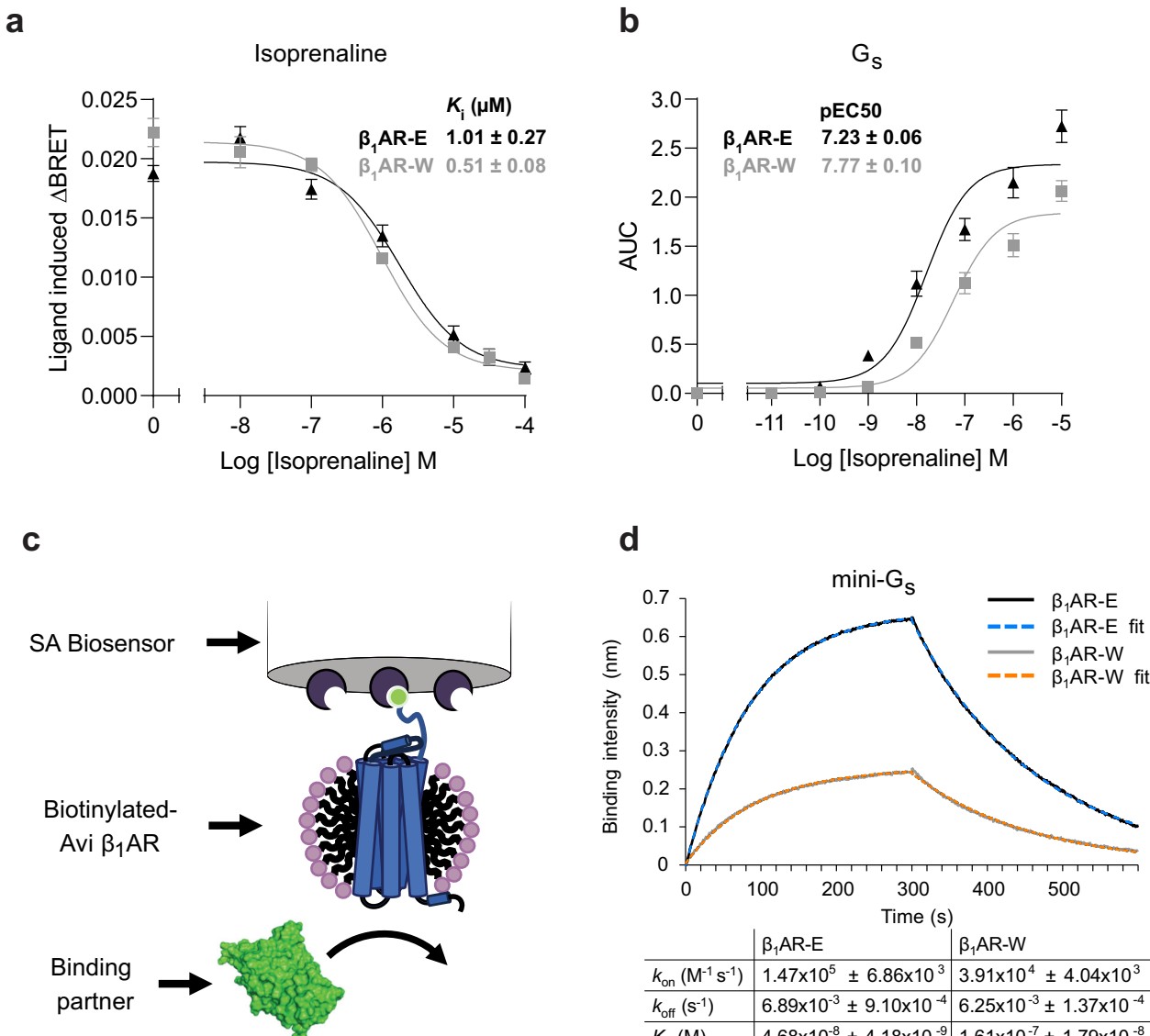

**Fig. 1 | *In cell* and in vitro validation of the β₁AR-E and β₁AR-W constructs.**
**a** Isoprenaline binding affinity of the β₁AR constructs expressed in HEK293T cells, measured using the nanoBRET signal between bound (S)-propranolol-red and either β₁AR-E (black) and β₁AR-W (grey) N-terminally conjugated with Nanoluc. A signal reduction is observed as increasing concentrations of isoprenaline compete for the orthosteric binding site. **b** In cell measurements of the dissociation of the G protein Gβγ subunit from Gα$_s$ using the BRET2-based TRUPATH G protein dissociation assay[46]. For both (**a**) and (**b**), data are mean ± SEM of $n = 3$ independent biological experiments each performed in duplicate. The responses were measured for varying isoprenaline concentrations to obtain estimates of potency for β₁AR-E (black) and β₁AR-W (grey). **c** Schematic of the BLI assay illustrating the immobilisation of receptor (blue) via a biotinylated N-terminus onto a streptavidin (dark purple) coated biosensor. In the current study the receptor is detergent micelle solubilised (pink head groups and black tails). Soluble binding partners such as mini-G (green) bind to the cytoplasmic pocket of the receptor leading to a measurable BLI response. **d** Example of BLI binding traces for isoprenaline-bound (500 μM) β₁AR-E (black) and β₁AR-W (grey) in the presence of mini-G$_s$ (62.5 nM for β₁AR-E, 125 nM for β₁AR-W). Fits to the data (β₁AR-E in blue) and (β₁AR-W in orange) are based on a 1:1 monophasic analysis. The calculated $K_D$, $k_{on}$ and $k_{off}$ values for both constructs are shown underneath the binding intensity plot. Values are averages of $n = 3$ individual repeats and standard deviations are shown.

indicated by our previous spectral assignments[20]. The isoprenaline-bound spectrum varied substantially from the corresponding spectrum of β₁AR-W (Supplementary Fig. 6a), which occupied a pre-active state, and looked similar to the previously recorded ternary complex spectrum of β₁AR-W bound to isoprenaline and Nb6B9[20] (Supplementary Fig. 6b). Therefore, the full agonist bound β₁AR-E occupied a global conformation considerably closer to the solution Nb6B9 ternary complex than β₁AR-W, which we concluded was representative of the solution active state.

In the receptor core, M223[5.54] and M296[6.41] showed marked shifts which indicated significant conformational rearrangements around the nearby highly conserved P[5.50]-I[3.40]-F[6.44] motif[50]. The latter has been

discussed to play a crucial role in signal transduction from the binding pocket and stabilisation of the active state[51]. A comparison of the structures of inactive β₁AR with the active full agonist and G protein bound β₁AR ternary complex showed conserved aromatic residues F299[6.44] and Y227[5.58] adopt orientations which would result in a strong shielding effect on M223[5.54] and deshielding effect on M296[6.41], respectively (Supplementary Fig. 6c, d). These changes were consistent with the ¹H shift perturbations we observed. Further, the large upfield ¹³C shift for M296[6.41] indicated a more gauche orientation of the $\chi^3$ angle in the agonist-bound state, in agreement with the ternary complex structure but also possibly resulted from changes in shielding[52] due to the changed orientation of F299[6.44]. Therefore,

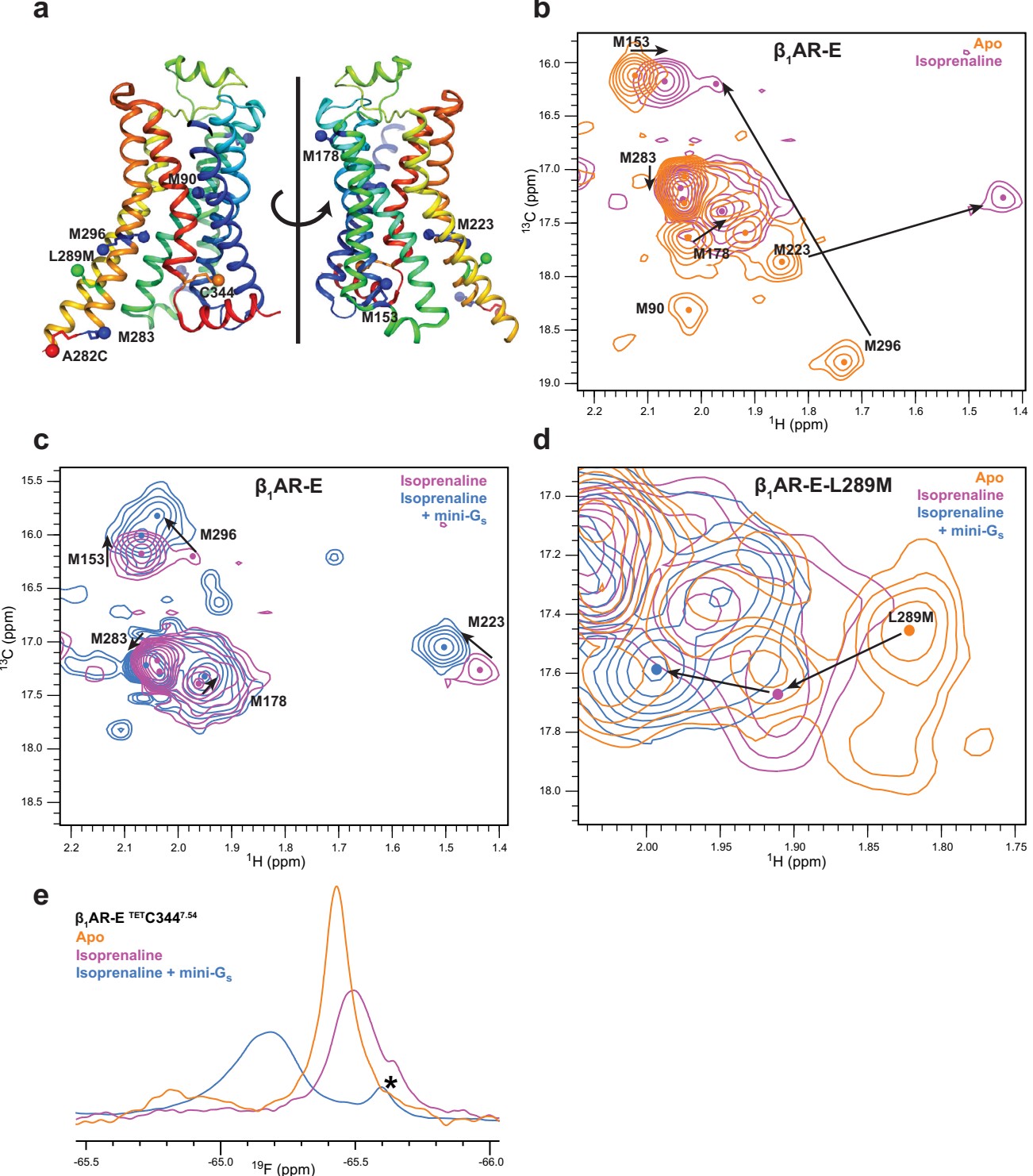

**Fig. 2 | NMR spectra of isoprenaline-bound β₁AR-E show the receptor populating an active state that differs in its structure from the ternary complex with mini-Gₛ. a** The positions of the assigned ¹H-¹³C and ¹⁹F NMR probes are mapped onto the structure of β₁AR in ternary complex with isoprenaline and activating nanobody (PDB: 6H7J) where they are indicated as coloured spheres: ¹³C methyl groups of the native methionine residues (blue), introduced methionine L289M[6.34] (green), ¹⁹F TET probe conjugated to the native cysteine C344[7.54] (orange) and the ¹⁹F BTFA probe conjugated to cysteine A282C[6.27] (red) (Figure made with PyMol V2.4.1). **b–d** Superposition of 2D ¹H-¹³C HMQC correlation spectra with assignments

indicated for β₁AR-E in the ligand free apo state (orange), in the active state bound to isoprenaline (purple) and in ternary complex bound to isoprenaline and mini-Gₛ (blue). The enlarged view in (**d**) shows the signal response of L289M[6.34] on TM6 upon stepwise receptor activation of β₁AR-E-L289M[6.34]. For guidance, sizeable changes in chemical shift upon activation are indicated by black arrows. The individual peak centres are indicated by dots. **e** 1D ¹⁹F NMR spectra of β₁AR-E labelled at ᵀᴱᵀC344[7.54] on TM7 showing the apo receptor (orange), the isoprenaline-bound active state (purple) and the ternary complex of β₁AR bound to isoprenaline and mini-Gₛ (blue). A known degradation peak is indicated by an asterisk.

changes to both M223[5.54] and M296[6.41] signals strongly indicated that in the solution isoprenaline-bound state, the core of the receptor approached the conformation observed in the ternary complex structure. Additionally, the upfield shift in [13]C for M223[5.54] indicated a greater degree of trans-gauche conformational exchange, and line broadening of both M223[5.54] and M296[6.41] suggesting that the adopted state of β₁AR-E remained dynamic on the μs-to-ms timescale and sampled different local environments. These observations implied an increase in the dynamics of the PIF motif region upon agonist binding, and a substantially more dynamic receptor core relative to the β₁AR-W pre-active state.

Effects of agonist binding also permeated to the extracellular and intracellular surfaces. M178[4.62] at the top of TM4 resolved into a single peak in the isoprenaline-bound β₁AR-E spectrum relative to two peaks in the apo and β₁AR-W pre-active states (Fig. 2b and Supplementary Fig. 6a), indicating reduced conformational variability as the receptor orthosteric binding pocket formed favourable interactions with the agonist. Also, the IL2 probe M153[34.57], which is distant from the ligand binding pocket (~26 Å), showed a more substantial upfield [1]H shift upon addition of isoprenaline (Fig. 2b) relative to β₁AR-W (Supplementary Fig. 6a), indicating greater activation[20] and allosteric conformational changes to either IL2 or the cytoplasmic tip of TM3 as the effect of agonist binding propagated to the intracellular region.

Significant rearrangement of TM6 is associated with class A GPCR ternary complex formation[53]. Therefore, we assessed conformational changes in this key region using the introduced probe L289M[6.34] which occupies a less solvent-accessible location between M283[6.28] and M296[6.41] (Fig. 2a). The equivalent mutation was previously introduced into β₂AR with only modest effects on ligand affinity[10] and the non-disruptive nature of the L289M[6.34] mutation was supported by the similarity of the native methionine signal positions between L289M[6.34] and L289[6.34] spectra, in the apo and isoprenaline-bound conditions (Supplementary Fig. 6e). We observed a pronounced [1]H downfield shift of L289M[6.34] upon isoprenaline addition, indicating a major conformational rearrangement of TM6 upon agonist binding (Fig. 2d). This peak change agreed with movement of L289[6.34] towards Y231[5.62] in the β₁AR-Gₛ ternary structure (PDB: 7JJO)[41] and the paramagnetic relaxation enhancement (PRE)-based β₂AR isoprenaline-bound structure (PDB: 6KR8)[25] where full agonist binding induces rotation of the lower half of TM6 (Supplementary Fig. 6f). Therefore, L289M[6.34] indicated TM6 underwent a significant conformational change in solution upon isoprenaline addition. In contrast, no such movement was observed in the static inactive cyanopindolol-bound or isoprenaline-bound receptor structures (PDB: 2YCY and 2Y03)[43,54]. The latter is in agreement with X-ray structures of full agonist-bound receptors which frequently show TM6 in an inactive (or inactive-like) conformation[55].

Corroborating information was obtained for the cytoplasmic tip of TM6 through 1D [19]F NMR in combination with single site [19]F-labelling at the previously studied A282C[6.27] position[19] (Supplementary Fig. 7a). The isoprenaline-bound spectrum showed a peak which could be deconvoluted into two components (Supplementary Fig. 7b). The major of the two signals was shifted downfield, indicating a greater solvent accessibility than the minor peak based on a correlation between solvent accessibility and chemical shift observed for β₁AR-W[19] and confirmed using Gd[3+] solvent PRE experiments (Supplementary Fig. 7c). These changes in solvent accessibility were consistent with helical rotation of TM6 to move A282[6.27] from behind TM5 into the cytoplasm (Supplementary Fig. 7d), which we hypothesised represented the active state observed with L289M[6.34]. Conversely, the minor, less accessible peak indicated a conformation in which A282[6.27] was still positioned behind TM5, likely representing the pre-active state. Relaxation dispersion (CPMG) experiments did not show exchange on the μs-to-ms timescale between the two states (Supplementary Fig. 7e). However, saturation transfer (CEST) [19]F NMR experiments demonstrated exchange between the active and pre-active states

(Supplementary Fig. 8) consistent with slow ms-to-s exchange and helical movements, agreeing with L289M[6.34] observations.

Additional information on the nearby NPxxY motif in TM7 and neighbouring IL4 was obtained using [19]F-TET labelled C344[7.54] (Supplementary Fig. 7a). This position has been examined previously in β₂AR with the E122W[3.41] mutation[56], and in our previous work on β₁AR-W which revealed the isoprenaline-bound [TET]C344[7.54] as a single μs-to-ms exchange broadened signal[19]. The corresponding probe in β₁AR-E resulted in an even broader peak (Fig. 2e) that could be deconvoluted into two components (Supplementary Fig. 7f). The more intense [TET]C344[7.54] upfield component was unique to β₁AR-E, whereas the weaker component matched the β₁AR-W construct peak[19]. Therefore, we concluded that full agonist-bound β₁AR-E predominantly adopted a previously unobserved conformation in the TM7/ICL4 region but also populated to a lesser extent the pre-active state. The observation of two conformations was similar to that made for extracellular signals of β₁AR using [1]H-[15]N NMR[18]. The more severe signal broadening in the 2D [1]H-[13]C NMR spectra likely prevented the observation of the minor pre-active state signals. Notably, the populations of the states was similar to those for A282C[BTFA,6.27], indicating an allosteric link between TM6 and TM7 in the pre-active/active state equilibrium. Furthermore, the upfield location for the [TET]C344[7.54] major peak indicated a more hydrophobic environment relative to the pre-active state[19]. Occlusion of Y[7.53] in a hydrophobic layer was observed in agonist-bound MD simulations of the β₂AR, adenosine A₂A (A₂AR), and κ-opioid receptors[57,58], and similarly found in the β₂AR PRE structure (PDB: 6KR8)[25]. Our data support a similar β₁AR-E active state conformation with the probe and adjacent Y343[7.53] in the hydrophobic layer, suggesting that the cytoplasmic binding cavity remains occluded in the full agonist bound state. We therefore concluded that whilst the β₁AR-E active state conformation was similar to ternary complex static structures around the PIF motif, TM6 underwent a characteristic rotation of its helix. However, in the absence of a cytoplasmic coupling partner, displacement of TM6 from the receptor core had not occurred thus retaining the hydrophobic layer around the NPxxY motif. In contrast, the minor pre-active state peak showed smaller rotation of TM6 and reduced TM7 rotation into the hydrophobic layer as we previously demonstrated[19].

## Conformational changes upon β₁AR-E ternary complex formation

After establishing the similarities and differences between the full agonist-bound solution active conformation of β₁AR and the ternary complex static structures, we questioned what further conformational changes were required to reach the solution ternary complex conformation. We therefore characterised receptor conformational changes upon transition from the isoprenaline-bound active state to the ternary complex by recording 2D [1]H-[13]C HMQC experiments of the β₁AR-E/isoprenaline/mini-Gₛ ternary complex. None of the NMR probes had direct interactions with proximal binding partner residues, thus changes to chemical shifts reported receptor conformational changes upon mini-Gₛ binding.

The ternary complex 2D [1]H-[13]C HMQC spectrum showed peak locations that were reminiscent of isoprenaline-bound positions (Fig. 2c), indicating the solution ternary complex receptor conformation was similar to the solution active state. Therefore, the major conformational changes involved in receptor activation occurred upon agonist binding rather than coupling to mini-Gₛ, in contrast to our previous studies using thermostabilised β₁AR-W with nanobody[20]. Nevertheless, notable smaller differences for all methionine probes were observed, indicating there were further minor conformational adjustments upon mini-Gₛ binding. Further, the generally greater peak intensities of the ternary complex spectrum showed that the ternary complex was less dynamic than the agonist-bound state (Supplementary Fig. 9a). Together, these observations suggested that whilst the

receptor bound to agonist was already primed in a conformation suitable to engage with mini-$G_s$, further structural adjustments were required to form the ternary complex, resulting in a more rigid receptor.

Positions of peaks corresponding to residues located in the extracellular and core transmembrane regions were only mildly affected by mini-$G_s$ binding. The change for M178[4.62] reflected the established allosteric contraction of the ligand binding pocket that accompanies ternary complex formation[18,59,60] which results in an increased affinity of the bound agonist[59,61]. The minor $^1H$ shift change for M223[5.54] indicated only a small adjustment relative to F299[6.44] in the PIF motif, confirming the active state was already similar in conformation to the solution ternary complex in this conserved region. The M223[5.54] $^{13}C$ shift indicated residual dynamics of the side chain via $\chi^3$ trans/gauche conformational exchange in agreement with published structures showing M223[5.54] with a range of $\chi^3$ dihedral angles. However, the peak intensity increased by 2.4-fold (Supplementary Fig. 9a), indicating that the region around M223[5.54], including the PIF motif, adopted a more rigid conformation than in the ligand-bound active state of $\beta_1$AR-E.

M296[6.41] also showed a subtle change in proton shift, likely in response to readjustments of the PIF motif and Y227[5.58] as it engaged with Y343[7.53] on TM7. The substantial 3.6-fold increase in peak intensity (Supplementary Fig. 9a) indicated that the TM5-TM6 interface around M296[6.41] was also less dynamic and thus stabilised by mini-$G_s$ binding.

In IL2, closer to the binding pocket, only a minor change in the $^1H$ dimension was observed for M153[34.57] which implied small differences in the surrounding IL2 conformation upon coupling to mini-$G_s$, consistent with structural comparisons (Supplementary Fig. 9b). Conversely, the M153[34.57] $^{13}C$ shift moved more appreciably towards a gauche conformation, suggesting reduced dynamics in the wider IL2 and TM3 cytoplasmic region in the presence of binding partner in agreement with overall receptor rigidity increases.

The L289M[6.34] and M283[6.28] probes, located on the cytoplasmic side of TM6, showed chemical shift changes more similar in magnitude to those observed upon isoprenaline addition to the apo receptor. This confirmed that the major conformational changes accompanying ternary complex formation concentrated at the cytoplasmic binding interface.

A downfield shift in $^1H$ of the L289M[6.34] probe (Fig. 2d) suggested movement towards the proximal side chain of Y231[5.62] and supported further opening of the cytoplasmic cavity via adjustments in TM6 orientation upon mini-G protein binding. This was consistent with the smaller L289[6.34]-Y231[5.62] distance observed in ternary complex (PDB: 7JJO)[41] relative to the agonist-bound $\beta_2$AR PRE structure (PDB: 6KR8)[25] (Supplementary Fig. 6f). M283[6.28] at the tip of TM6 appeared as a sharp peak indicating fast local dynamics consistent with a solvent exposed environment. The $^{13}C$ chemical shift remained in the rapid trans/gauche exchange shift range, supporting a high flexibility of this region and in agreement with the ternary structure (PDB: 7JJO)[41] in which M283[6.28] was unresolved. For the adjacent A282C[BTFA,6.27] $^{19}F$ probe, addition of mini-$G_s$ resulted in a single sharp signal that was downfield shifted relative to the ligand-bound states (Supplementary Fig. 9d, e). The smaller linewidth relative to the agonist-bound state indicated faster dynamics which agreed with M283[6.28], likely due to increased flexibility and solvent accessibility. Together, M283[6.28] and A282C[BTFA,6.27] suggested an increase to solvent exposure and flexibility in the TM6 cytoplasmic tip upon binding partner coupling, consistent with additional displacement farther into the cytoplasm.

Addition of mini-$G_s$ to isoprenaline-bound $^{TET}C344^{7.54}$ resulted in a $^{19}F$ downfield shift (Fig. 2e) consistent with a transition to a more hydrophilic solvent exposed environment. Therefore, this shift supported breaking of the hydrophobic layer and extension of Y343[7.53] into the cytoplasmic cavity. This suggests that outward displacement of TM6, and hence opening of the cytoplasmic cavity, only occurs

upon complex formation. Consequently, the $^{TET}C344^{7.54}$ ternary complex spectrum further supported our proposed agonist-bound active state conformation. Interestingly, the $^{TET}C344^{7.54}$ signal deconvoluted as two components with minor differences in shifts but differing linewidths (Supplementary Fig. 9f, see Supplementary Fig. 10 for single component deconvolutions). This observation, coupled with the lack of proximity to either the agonist or mini-$G_s$ binding surfaces, suggested that the receptor region near IL4 sampled two subtly different environments in the mini-$G_s$ complex, and implied IL4 retains conformational dynamics in ternary complexes.

We confirmed further $\beta_1$AR-E conformational changes were present upon ternary complex formation with the active-state stabilising nanobody Nb6B9. Although generally similar, discrepancies in peak positions and intensities relative to the mini-$G_s$ ternary complex were prominent in the cytoplasmic IL2, TM6, and TM7/IL4 regions (Supplementary Fig. 11). This variation implied differences in local receptor conformation and complex dynamics, emphasising the plasticity and conformational adaptability of $\beta_1$AR to accommodate structurally different coupling partners.

Overall, our NMR data suggested that further conformational changes, including altering of the cytoplasmic binding cavity through TM6 displacement, were necessary to reach the final ternary complex conformational state in solution. We concluded that full agonist ternary complex formation progressed mainly via conformational selection of the active state followed by smaller amounts of induced fit upon mini-$G_s$ binding.

## Ternary $G_s$ complex formation via the pre-active state

In our previous studies, isoprenaline and partial agonist-bound $\beta_1$AR-W occupied the inactive/pre-active state equilibrium[19,20]. Likewise, based on our $^1H$-$^{13}C$ and $^{19}F$ NMR data we found partial agonist (xamoterol, intermediate efficacy; salbutamol, high efficacy)[62] bound $\beta_1$AR-E populated the same inactive/pre-active state equilibrium, with no evidence of active state populations (Fig. 3a, c–e, Supplementary Fig. 12, Supplementary Fig. 13a). This observation suggested partial agonist-bound receptor was unable to overcome the energy barrier to occupy the active state. Consequently, we utilised this to investigate the aptitude of the pre-active state to interact with mini-$G_s$ to form ternary complexes and whether intrinsic differences between the pre-active state and the active state would result in conformational differences in the respective ternary complexes with mini-$G_s$.

Salbutamol and xamoterol had decreased potency in TRUPATH assays relative to isoprenaline, indicating reduced signalling through $\beta_1$AR-E (Supplementary Fig. 14a, b). Similarly, BLI assays with salbutamol- and xamoterol-bound $\beta_1$AR-E demonstrated reduced mini-$G_s$ binding affinities relative to the equivalent isoprenaline assays (Fig. 3b, isoprenaline with salbutamol: $p = 0.0018$; isoprenaline with xamoterol: $p = 0.0052$). The increased $k_{off}$ rates (isoprenaline with salbutamol: $p = 0.0021$; isoprenaline with xamoterol: $p = 0.003073$) indicated partial agonist ternary complexes were slightly less stable than for those with full agonist (Supplementary Fig. 14c, Supplementary Table 3), agreeing with assays using fluorescent BODIPY-GTPγS binding to complexes in other studies[63]. However, as with isoprenaline-bound $\beta_1$AR-W, the lower $k_{on}$ rates relative to isoprenaline-bound $\beta_1$AR-E (isoprenaline with salbutamol: $p = 0.00077$; isoprenaline with xamoterol: $p = 0.00075$) were the major contributor to differences in $K_D$ (Supplementary Fig. 14c, Supplementary Table 3) and therefore suggested the more occluded pre-active state had a higher energy barrier to ternary complex formation compared to the active state.

We also conducted BLI assays with partial agonist-bound $\beta_1$AR-W, which showed only minor decreases in mini-$G_s$ binding affinity compared to $\beta_1$AR-E (Fig. 3b) mainly influenced by the modestly higher $k_{off}$ rates, indicating similar partial agonist-bound receptor pre-active state populations. Following the kinetic investigations of the pre-active state to mini-$G_s$ binding, we characterised the pre-active state induced

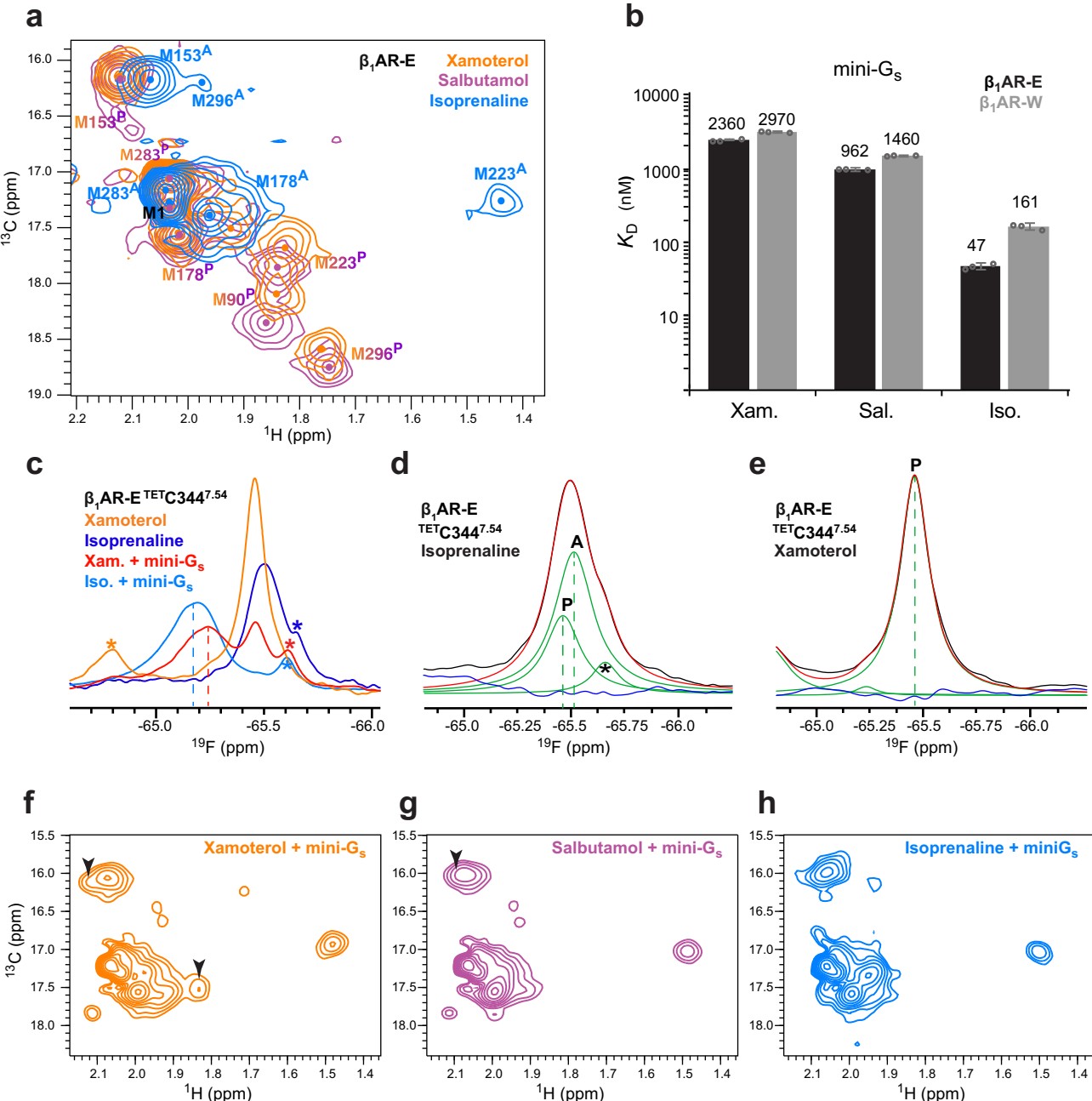

**Fig. 3 | Pre-active and active state populations of β₁AR are agonist-dependent and affect ternary complex formation with mini-Gₛ. a** Overlay of 2D ¹H-¹³C HMQC spectra of ligand-bound β₁AR-E with the full agonist isoprenaline (blue) or the partial agonists xamoterol (intermediate efficacy) (orange) or salbutamol (high efficacy) (magenta). Assignments of peaks related to the pre-active state are indicated by (ᴾ) while those related to the active state are indicated by (ᴬ). **b** Affinity of β₁AR (β₁AR-E - black, β₁AR-W – grey) ternary complex formation with mini-Gₛ in the presence of full (isoprenaline) or partial agonists (xamoterol, salbutamol) measured by BLI. Values are the averages of $n = 3$ individual repeats, and error bars indicate the SD of these replicates. See Methods and Supplementary Methods for details. **c** Overlay of 1D ¹⁹F NMR spectra of ᵀᴱᵀC344⁷·⁵⁴ of β₁AR-E ligand-bound to either isoprenaline (dark blue) or xamoterol (orange), or in ternary complex with mini-Gₛ (8 molar equivalents) and xamoterol (red) or mini-Gₛ (2 molar equivalents) and isoprenaline (light blue). Differences in the signal positions of the ternary complexes indicate ligand dependent variations in the TM7/IL4 region. Vertical

dotted lines indicate the signal position of the ternary complexes. Known degradation products are marked by an asterisk. **d, e** ¹⁹F NMR signal of ᵀᴱᵀC344⁷·⁵⁴ for β₁AR-E bound to isoprenaline (**d**) or xamoterol (**e**) reveals the existence of two receptor populations (active state A and pre-active state P) when bound to isoprenaline, but only one with xamoterol. The recorded spectrum is shown in black, with the deconvoluted signal components shown in green. The reconstructed spectrum is shown in red with the residual trace shown in blue. Peak positions are indicated by vertical dotted lines and labelled according to the active state (A) or the pre-active state (P). The asterisk indicates a known degradation product. Side-by-side comparison of ¹H-¹³C spectra of β₁AR-E-L289M⁶·³⁴ in ternary complex with mini-Gₛ (8 molar equivalents) in the presence of xamoterol (**f**), salbutamol (**g**) or isoprenaline (**h**), respectively. The black arrow heads indicate positions of residual ligand-bound receptor signals that are present due to the reduced affinity of the partial agonist-bound ternary complexes for mini-Gₛ.

ternary complex conformation. The $^1$H-$^{13}$C NMR spectral fingerprint of isoprenaline-bound $\beta_1$AR-W in complex with mini-G$_s$ was very similar to the ternary complex with $\beta_1$AR-E. Although the structural changes in the vicinity of the W130E$^{3.41}$ mutation affected nearby residues in the core region (M223$^{5.54}$, M296$^{6.41}$), the cytoplasmic region (L289M$^{6.34}$, M283$^{6.28}$, M153$^{34.57}$) remained very similar (Supplementary Fig. 15a, b), indicating that the final ternary complex conformation of $\beta_1$AR was mostly independent of the initial engagement of mini-G$_s$ with the active or pre-active state of the receptor.

In $^1$H-$^{13}$C HMQC spectra of partial agonist bound $\beta_1$AR-E ternary complexes with mini-G$_s$ (Fig. 3f, g) residual ligand-bound peaks were still present consistent with the lower mini-G$_s$ binding affinities. However, the methyl probe positions corresponding to the ternary complexes were reminiscent of the equivalent peaks of the complex with isoprenaline (Fig. 3h). These data indicated a similar global receptor conformation in solution for mini-G$_s$ ternary complexes irrespective of coupling to receptor bound to full agonist or partial agonist. Notably, the chemical shifts of probes on the cytoplasmic region of TM6, L289M$^{6.34}$ and M283$^{6.28}$, were comparable between pre-active and active state induced ternary complexes (Supplementary Fig. 15c, d).

$^{19}$F NMR saturation transfer experiments with A282C$^{BTFA,6.27}$ confirmed that the ternary complex TM6 signal exchanges with the pre-active state signal (Supplementary Fig. 13b). Additionally, ternary complexes showed minimal variation between the mini-G$_s$ complexes with isoprenaline-bound $\beta_1$AR-E and isoprenaline-bound $\beta_1$AR-W, or with either salbutamol- or xamoterol-bound $\beta_1$AR-E (Supplementary Fig. 16a). Therefore, we concluded the TM6 conformation and degree of outward displacement was conserved between the ternary complexes, likely reflecting stabilisation by the binding partner.

However, other NMR probes did display minor differences. M178$^{4.62}$ showed adjustments to the ligand binding pocket whilst M153$^{34.57}$ displayed changes to $^1$H shifts reporting on small differences to the IL2 conformation (Fig. 3f–h, Supplementary Fig. 15c–e). These changes were consistent with our previous work which showed these residues to be a sensitive measure of receptor activation and ligand efficacy[20].

Slight changes to chemical shifts for M223$^{5.54}$ and M296$^{6.41}$ in the receptor transmembrane core (Supplementary Fig. 15e, f) indicated partial agonists influenced the conformational exchange of the PIF motif and TM5-TM6 interactions.

Relative to their equivalent ligand-bound $^{19}$F NMR spectra, the salbutamol- or xamoterol-bound $\beta_1$AR-E and isoprenaline-bound $\beta_1$AR-W ternary complexes showed downfield shifted peaks for $^{TET}$C344$^{7.54}$ (Fig. 3c) confirming the anticipated breaking of the hydrophobic layer in the NPxxY and IL4 region seen during ternary complex formation. In agreement with the different orientations of Y343$^{7.53}$ as stipulated for the ternary complex formed via the $\beta_1$AR-E active state, the signals deconvoluted into the same two peak components (Supplementary Fig. 16b–d). However, the complexes formed via the pre-active state preferably populated the more upfield shifted signal component, in contrast to the complexes formed via the active state. Consequently, we hypothesised ternary formation via the active state resulted in greater population of a fully extended Y343$^{7.53}$ conformation, maintaining the water-mediated hydrogen bond[3] between Y$^{5.58}$ and Y$^{7.53}$. The greater stability of the hydrogen bond agreed with the higher stability of isoprenaline-bound $\beta_1$AR-E shown via BLI (Supplementary Fig. 14c). Conversely, complex formed via the pre-active state shifted equilibria towards a conformation with reduced extension of Y343$^{7.53}$ without the hydrogen bond, thus reducing complex stability. Variation of Y$^{5.58}$ was also consistent with the subtle changes observed for M223$^{5.54}$ and M296$^{6.41}$.

Together, our observations suggested ternary complex formation via the pre-active state required a greater induced fit contribution than via the active state but resulted in similar conformations, though partial agonists still imparted minor differences in the receptor core and binding interface periphery.

## G protein binding kinetics convey subtype selectivity

Having characterised the primary coupling of G$_s$ to $\beta_1$AR, we endeavoured to study the kinetics and dynamics of secondary coupling of non-canonical binding partners using the methodologies we established for G$_s$. We assessed functional activation of G$_q$ and the G$_{i/o}$ members G$_{i1}$ and G$_{o1}$ by $\beta_1$AR using the aforementioned BRET2-based TRUPATH assay[46]. Concentration-response curves were generated which inferred functional coupling to both $\beta_1$AR-W and $\beta_1$AR-E for G$_{i1}$, G$_{o1}$ and G$_q$, with reduced potencies for the $\beta_1$AR-W ($p = 0.0041$, $p = 0.029$, $p = 0.097$ respectively) (Fig. 4b). The changes in potencies scaled similarly for each binding partner between the two receptor constructs, which implied receptor thermostabilisation had little effect on G protein subfamily selectivity. Only a minor preference towards G$_s$ activation was observed, similar to observations for $\beta_2$AR[46].

Similar to mini-G$_s$, minimalised forms of other G proteins have been established[48]. We measured binding affinities for these proteins by BLI (Supplementary Fig. 17, Supplementary Table 4). The G$_{i1}$ equivalent, mini-G$_{i1}$, had poor thermal stability which prevented reliable quantitative analysis of binding kinetics. Since G$_{o1}$ showed similar activation to G$_{i1}$ in the TRUPATH assays, and these proteins have 73% identity (Supplementary Fig. 18a), the engineered mini-G$_{o1}$ was substituted for mini-G$_{i1}$. Additionally, the chimeric mini-G$_{s/i}$ (a chimera of mini-G$_s$ with the receptor-interacting residues of the α5 helix replaced by G$_{i1}$ residues (Supplementary Fig. 18b) and shown to have G$_{i1}$-like characteristics[48]), and equivalent mini-G$_{s/q}$ chimera were also characterised. The two chimeras were established on the observation that the majority of residues interacting with the receptor were in α5 and thus were proposed as more stable mimetics of G$_{i1}$ and G$_q$, respectively[48].

Assessment of complex formation with isoprenaline-bound $\beta_1$AR-E showed non-G$_s$ mini-G mimetics had significantly ($p < 0.05$) reduced affinities, between 5–30 fold, relative to the primary binding partner mimetic mini-G$_s$ (Fig. 3c, Supplementary Table 5, see Supplementary Table 6 for $p$ values). These reductions in affinities were predominantly the result of slower $k_{on}$ rates. Notably, the similarity in $k_{off}$ rates between the mini-G proteins implied that the complexes were similarly stable once formed. Accordingly, the higher selectivity of $\beta_1$AR for G$_s$ was reflected in its enhanced ability to associate with this binding partner, rather than the stability of the final ternary complex conformation.

The affinity for mini-G$_{o1}$ (1.34 μM) differed 28.5-fold from mini-G$_s$ (47 nM), with its $k_{on}$ rate reduced by 50-fold. Mini-G$_{s/i}$ showed a 23-fold reduction in affinity (1.09 μM), with a 40-fold lower on-rate, indicating the α5 helix residues contributed significantly ($p = 0.00075$, Supplementary Table 6) to the differences in $k_{on}$ rates and affinities. This agreed with the observation that 80% of the contacts between the two proteins in the ternary complex involved this helix[27]. The $K_D$ for mini-G$_{s/q}$ was lower (0.24 μM), more similar to the mini-G$_s$ complex though still ~5-fold greater. This further supported alteration of amino acids in the α5 helix as sufficient to modulate coupling to receptor, given the two chimeras share the same G$_s$ Ras domain.

Equivalent assays with the $\beta_1$AR-W construct showed that the order of selectivity reflected in the affinities and $k_{on}$ rates of the different mini-G proteins for $\beta_1$AR-E were maintained (Fig. 4c), in agreement with the TRUPATH assay. However, binding was less tight compared to $\beta_1$AR-E, which again was primarily the result of reductions in $k_{on}$ rates, consistent with the more occluded pre-active state populated by $\beta_1$AR-W.

Based on the kinetics of complex formation we hypothesised that higher rates of association for the G$_s$ protein facilitated selectivity by outcompeting association of other G protein family members.

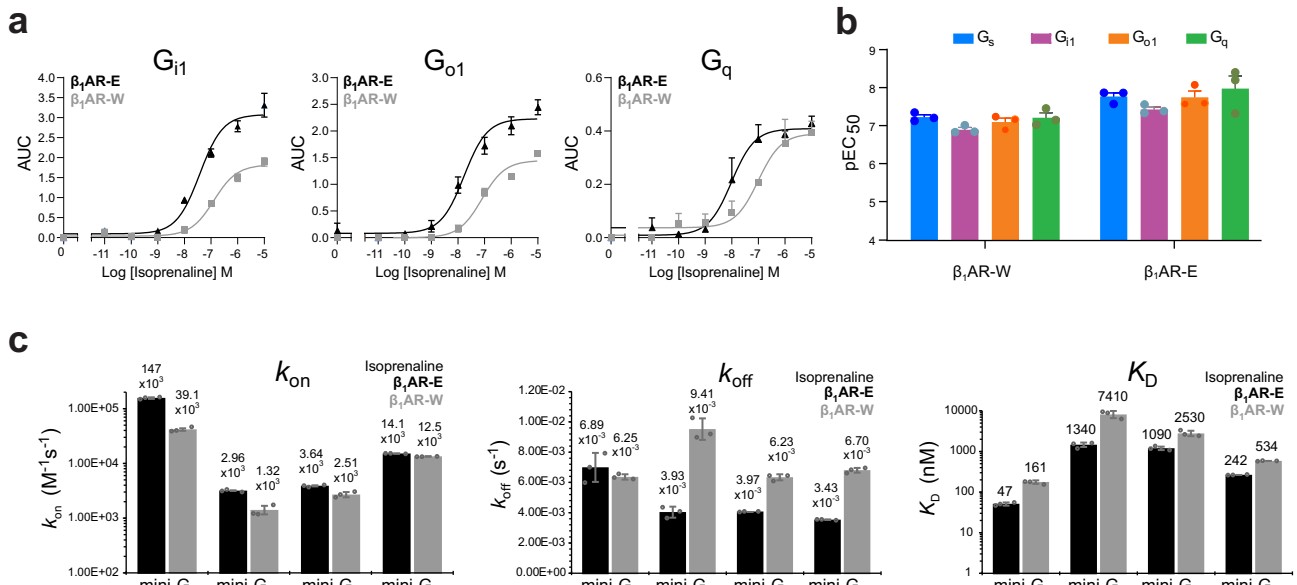

**Fig. 4 | Ternary complex formation of β₁AR isoprenaline-bound with non-canonical G proteins (secondary coupling). a** In cell measurement of the dissociation of the Gβγ G protein subunit from Gα from different families using the BRET2-based TRUPATH assay for G protein activation. Signal was measured for varying isoprenaline concentrations to obtain concentration-response curves for β₁AR-E (black) and β₁AR-W (grey). **b** pEC₅₀ values for isoprenaline bound β₁AR measured using the TRUPATH assay. Values were obtained for both β₁AR-E and β₁AR-W constructs signalling via Gₛ, Gᵢ₁, Gₒ₁ and G_q. For both (**a**) and (**b**), data are mean ± SEM of *n* = 3 independent biological experiments each performed in duplicate. **c** In vitro BLI binding kinetics and affinity data for different mini-G proteins binding to detergent solubilised avi-β₁AR-E (black) and avi-β₁AR-W (grey). The mini-Gₛ shows a dramatically increased $k_{on}$ rate compared to the other proteins. Whilst the $k_{off}$ rates are more comparable, the significant differences in $k_{on}$ are the major contributor to the higher affinity of the mini-Gₛ ternary complex. See Supplementary Table 6 for exact *p* values. In brief, all $k_{on}$ values for β₁AR-E (log scale)

are significantly different from each other (*p* < 0.05) except those for mini-Gₒ₁ and mini-Gₛ/ᵢ. None of the $k_{off}$ values for β₁AR-E are significantly different from each other (*p* < 0.05) except those for mini-Gₛ/_q and mini-Gₛ/ᵢ. All $K_D$ values (log scale) for β₁AR-E are significantly different from each other (*p* < 0.05) except between mini-Gₒ₁ and mini-Gₛ/ᵢ / mini-Gₛ/_q. Statistical significance was calculated with unpaired two-tailed t-tests, with a Bonferroni correction to the significance level for multiple comparisons. All parameters between β₁AR-E and β₁AR-W for each binding partner are significantly different (*p* < 0.05) aside from $k_{off}$ rates for mini-Gₛ. See Supplementary Table 7 for exact *p* values comparing β₁AR-E and β₁AR-W kinetic parameters. Statistical significance was calculated with unpaired two-tailed *t*-tests. For convenience, the individual values for $k_{on}$, $k_{off}$ and $K_D$ are indicated above the bars in the charts. Values are the averages of *n* = 3 individual repeats, and error bars indicate the SD of these replicates (Supplementary Table 5). See Methods and Supplementary Methods for details.

## Local ternary complex plasticity between G protein families

We assessed whether the observed differences in the kinetics of ternary complex formation were reflected in the solution structures through investigation of β₁AR in complex with the corresponding mini-G proteins. Similar to our investigations with mini-Gₛ, we recorded ¹H-¹³C HMQC experiments with isoprenaline-bound β₁AR-E in the presence of mini-Gₒ₁ (Fig. 5a) and mini-Gᵢ₁. Although mini-Gᵢ₁ experiments were limited to shorter experiments even at a reduced temperature, due to the reduced mini-G protein stability, the data quality was sufficient to confirm the overall spectral features. We found a high overall similarity of the spectra between the mini-Gᵢ₁ and mini-Gₒ₁ complexes (Fig. 5b) and given the similar behaviour of Gᵢ₁ and Gₒ₁ in functional data (Fig. 4), we therefore used the more stable mini-Gₒ₁ as a mimetic of the Gᵢ/ₒ family.

Given the lower binding affinities for G proteins associated with secondary coupling, we differentiated between signal broadening due to mini-G binding kinetics from broadening due to conformational exchange in the ternary complex, which is independent of mini-G concentration, by obtaining ternary complex spectra with 2 and 8 molar equivalents of mini-G protein. At the two mini-G concentrations we found no indication of binding kinetic-induced changes in the spectra (Supplementary Fig. 18c) and no residual uncomplexed active state receptor peaks were resolvable, suggesting β₁AR-E was close to saturation with mini-G in all complexes.

The overall appearance of the mini-Gₒ₁ ternary complex spectra was similar to the mini-Gₛ spectrum (Fig. 5a), suggesting comparable overall ternary complex receptor conformations and similar

interactions between receptor and binding partner. The negligible differences for M178⁴·⁶² showed that the extracellular side was unaffected by G protein subtype. Interestingly, the shift and intensity changes of M223⁵·⁵⁴ were minor when comparing mini-Gₛ and mini-Gₒ₁ complexes, indicating that the proximal PIF motif showed no significant changes. Similar ¹H shifts for M296⁶·⁴¹ corroborated this, suggesting F299⁶·⁴⁴ adopted comparable conformations between complexes. Therefore, the similarity of the PIF conformations indicated that the activation level of β₁AR was not altered between primary and secondary coupling. Notably, the similar ¹H shifts of L289M⁶·³⁴ on TM6 indicated comparable proximities to Y231⁵·⁶² on TM5, thus the outward displacement of TM6 did not differ substantially between the mini-Gₛ and non-canonical mini-G protein complexes. This agreed with the β₁AR cryo-EM structures in complex with heterotrimeric Gₛ and Gᵢ (PDB: 7JJO and 7S0F, respectively)[27,41] and other GPCR structural studies with multiple G proteins[35,64] but contrasted with earlier reports that found varying levels of TM6 opening to accommodate Gₛ or Gᵢ when comparing the structures of complexes with different receptors[31-34]. The signal intensities of corresponding methionine residues showed little variation between the complexes suggesting that the local and global dynamics of the receptor were similarly affected in the presence of the different G proteins (Supplementary Fig. 18d).

However, there were some small differences in peak positions in probes located across the receptor, which indicated subtle variations in the global receptor conformations. In complex with mini-Gₛ, the residues M296⁶·⁴¹ (Fig. 5c), L289M⁶·³⁴ (Fig. 5d), and M283⁶·²⁸ (Fig. 5a)

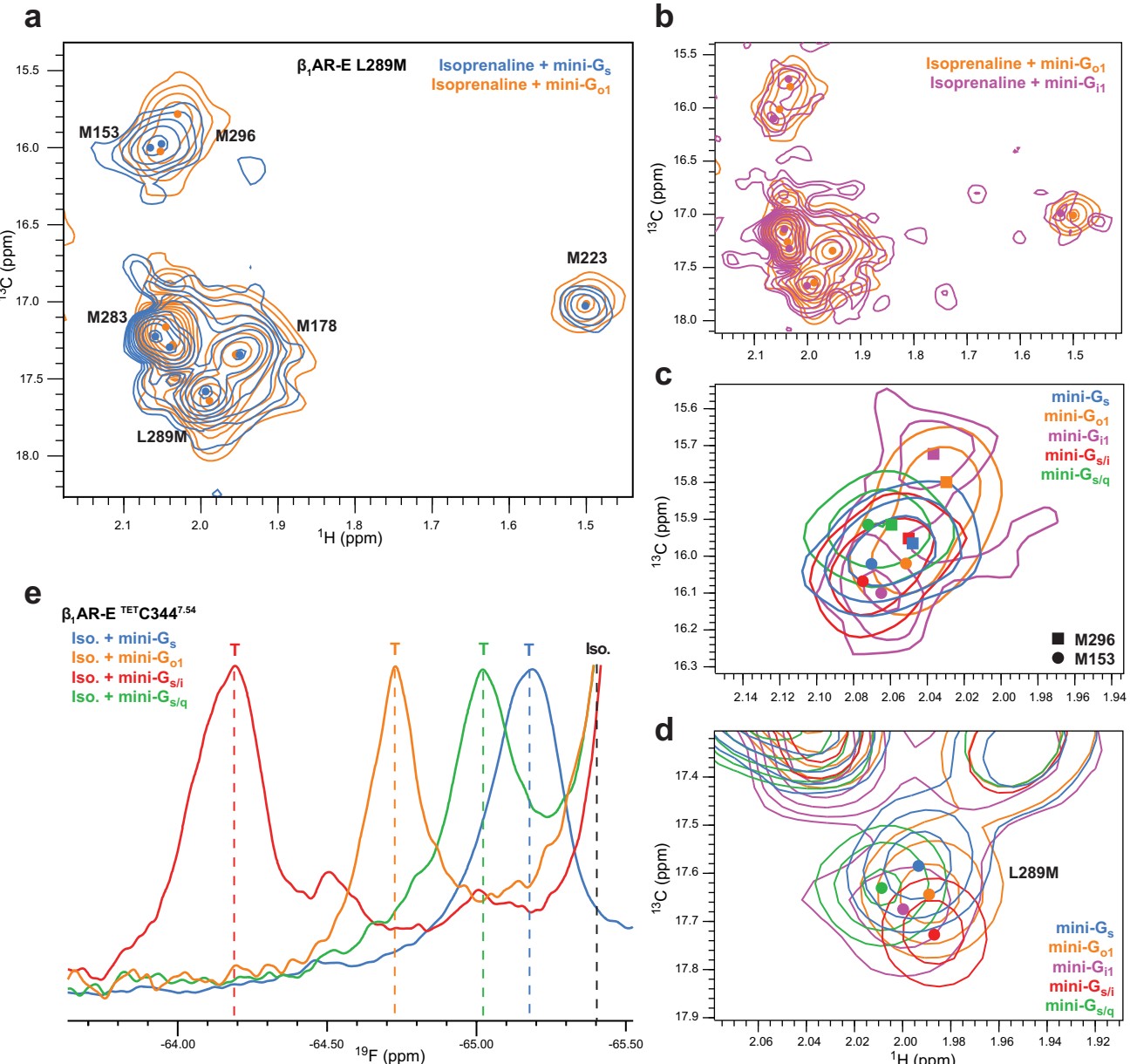

**Fig. 5 | NMR structural comparison of isoprenaline-bound $\beta_1$AR ternary complexes with canonical and non-canonical G proteins. a** Spectral overlay of $\beta_1$AR-E-L289M[6.34] in complex with mini-$G_s$ (blue) or mini-$G_{o1}$ (orange). **b** Similar comparison of $G_{i/o}$ family complexes with $\beta_1$AR-E-L289M[6.34] bound to mini-$G_{i1}$ (magenta) or mini-$G_{o1}$ (orange). The global resemblance of the NMR structural fingerprints provides evidence of strong structural similarity amongst the ternary complexes. However, local differences become obvious in close-ups of specific regions of the receptor (**c**, **d**) in complex with mini-$G_s$ (blue), mini-$G_{o1}$ (orange), mini-$G_{i1}$ (magenta) or the chimera proteins mini-$G_{s/i}$ (red), mini-$G_{s/q}$ (green): **c** focus on the region with M153[34.57] in IL2 and M296[6.41] on TM6; **d** close-up of L289M[6.34] on TM6. With the exception of the less stable mini-$G_{i1}$ (8 molar equivalents) all mini-G proteins were added at 2 molar equivalents. **e** 1D [19]F NMR spectra of $\beta_1$AR-E [TET]C344[7.54] in ternary complex with isoprenaline and either mini-$G_s$ (blue), mini-$G_{o1}$ (orange), mini-$G_{s/i}$ (red) or mini-$G_{s/q}$ (green) reveal differences in the TM7/IL4 region. The signal positions related to the ternary complexes (T) and agonist-only bound receptor (Iso.) are indicated by lines.

positioned along the lower cytoplasmic half of TM6 showed [13]C chemical shift values that were indicative of increased $\chi^3$ trans-gauche exchange of the methionine side chains relative to the mini-$G_{o1}$ and mini-$G_{i1}$ complexes, which adopted slightly more fixed trans or gauche conformers. Consequently, these probes suggested coupling to mini-$G_s$ (primary coupling) marginally reduced interactions between TM5 and TM6 along the helix interface, relative to coupling of non-canonical mini-G proteins (secondary coupling). Although these observations correlated with G protein selectivity, the small magnitude of the changes were incompatible with the large kinetic variation observed between mini-G proteins in our binding assays and therefore likely did not relate to selectivity.

Subtle conformational variation was also observed in IL2 as reported by [1]H shift differences for M153[34.57] (Fig. 5c) between mini-$G_s$ and mini-$G_{o1}$ complexes. These shifts were distinct from the isoprenaline-bound position of M153, in agreement with IL2 representing a key interacting region on the receptor. Other recent studies examining $G_s$ and $G_i$ binding to $\beta_2$AR[28,29] and muscarinic acetylcholine receptor type 2[28] have also noted variation in the IL2 region between different G protein family ternary complexes.

We observed the most substantial differences between binding partners in the TM7 and IL4 region in [19]F NMR experiments of [TET]C344[7.54] with mini-$G_s$ and mini-$G_{o1}$ complexes (Fig. 5e). The mini-$G_{o1}$ ternary complex peak was shifted substantially more downfield than

mini-G$_s$ relative to the ligand-bound peaks, consistent with an increased solvent accessibility[19] and suggesting that the NPxxY and IL4 environment was modulated by the presence of the different mini-G proteins. This indicated reduced contacts between the mini-G$_{o1}$ C-terminus and IL4 region relative to mini-G$_s$, possibly due to a less productive engagement of the two proteins during complex formation in this region.

We further recorded spectra with chimeric mini-G$_{s/i}$ and mini-G$_{s/q}$ proteins to examine how mutations in the α5 helix affected receptor conformation. Again, the general spectral features of the complexes were similar to complexes with mini-G$_s$ proteins (Supplementary Fig. 18e), indicating a similar overall structure of the complexes irrespective of the α5 helix identity. However, small variations were still present relative to mini-G$_s$ protein that extended across the entire receptor (Fig. 5d, e). Evidently, only a few mutations in the α5 helix were sufficient to affect the entire interaction interface, including regions around the binding cavity such as IL2, TM6, and TM7. Probes in the receptor core region and tip of TM6 more clearly showed specific influences of both the Ras domain identity and α5 helix on receptor conformations in ternary complexes (Fig. 5c, Supplementary Fig. 18f).

Together, our NMR data suggested that the global conformations of β$_1$AR in complex with G proteins from different families were highly similar. The small local conformational variations of the receptor in the different complexes seemed indicative of an intrinsic adaptability of β$_1$AR to adjust to different binding partners rather than suggestive of a role in G protein selectivity.

## Discussion

A rapidly increasing number of GPCR structures in complex with different G proteins are becoming available, however, a mechanistic understanding of G protein selectivity remains largely uncertain. We investigated the conformational signature of β$_1$AR in solution using $^1$H-$^{13}$C and $^{19}$F NMR spectroscopy in combination with position-selective isotope labelling to provide insights into the dynamic aspects of ternary complex formation. We characterised the receptor conformation in agonist-bound active state and in ternary complexes coupled to several Gα protein analogues from different families, and also the kinetics of G protein binding by BLI.

Our NMR data demonstrated the full agonist-bound β$_1$AR-E active state shared broad structural features with cryo-EM structures of G protein ternary complexes, but notably lacked displacement of the cytoplasmic side of TM6 away from the receptor core. This full agonist-bound conformation was reminiscent of the β$_2$AR PRE solution structure (PDB: 6KR8)[25], perhaps suggesting general structural characteristics for the active state conformation within the adrenergic receptor family and possibly the class A GPCRs in general. We found β$_1$AR-E also populated the pre-active state alongside the active state, as previously observed for a range of GPCRs[9,11,18,21,60,65–69]. The pre-active state conformation agreed with previous proposals[18–20,25] and was more reminiscent of static agonist-bound β$_1$AR crystal structures[43], suggesting these static structures represent the solution pre-active state. Interestingly, the energy barrier between the pre-active and active states could only be overcome through the allosteric interaction network stimulated by full agonist binding to β$_1$AR-E, whereas partial agonist bound β$_1$AR-E occupied the inactive/pre-active state equilibrium (Fig. 3).

Spectral changes upon mini-G$_s$ coupling to agonist-bound receptor suggested further conformational readjustments took place upon ternary complex formation primarily affecting the receptor core and cytoplasmic regions, including TM6 displacement and characteristic breaking of the hydrophobic layer[58] (Fig. 2). Our observations in solution are consistent with the increased TM6 displacement found in a single-molecule FRET comparison between agonist- and G protein-bound β$_2$AR[14]. The comparatively modest size of the conformational changes suggested engagement of the Gα binding partner through a

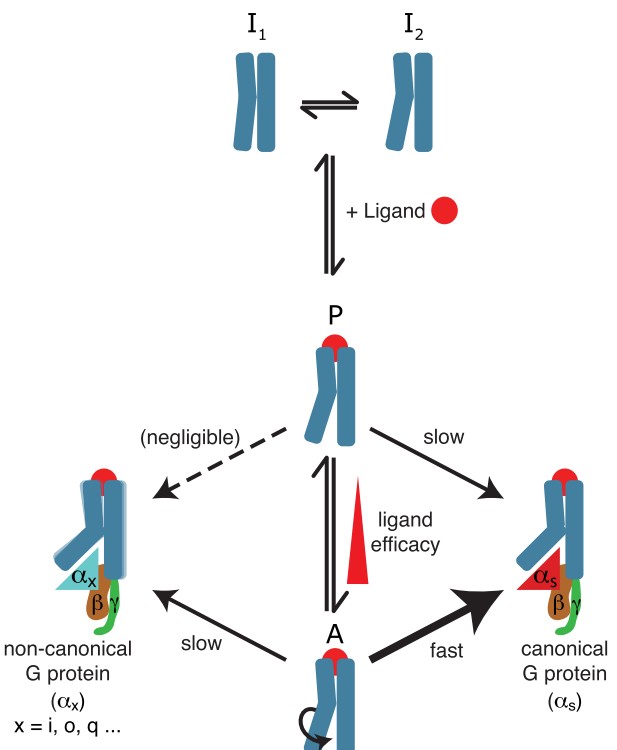

**Fig. 6 | Schematic illustration of the kinetic aspects of ternary complex formation contributing to G protein selectivity.** Agonist binding favourably shifts efficacy-dependent conformational equilibria between multiple states (inactive I$_1$, I$_2$; pre-active P; active A) of a GPCR towards the pre-active and active states. The active state (A) conformation shows a characteristic rotation of the cytoplasmic half of TM6, that promotes interaction with suitable binding partners. G protein interaction with the active or the pre-active state of the receptor leads to ternary complex formation following the displacement of TM6 from the receptor core and opening of the cytoplasmic binding cavity. The rate at which the ternary complexes form depends strongly on the family of G protein and the state the receptor is in. Favourable interactions between the Gα domain of the canonical binding partner and the active state of the GPCR result in substantially faster complex formation, outcompeting other non-canonical G proteins and contributing to selectivity. Only subtle variation in receptor conformation can be observed amongst the complexes with G proteins from different families.

conformational selection step that requires residual adjustments, in agreement with a recent study of adenosine A$_{2A}$R[24]. We also observed ternary complex formation with G$_s$ for the pre-active receptor state, with β$_1$AR bound to partial agonist (Fig. 3). Crucially, the receptor conformations in ternary complexes were generally similar irrespective of mini-G$_s$ initially binding to the pre-active state or the active state in solution. However, G protein binding to the pre-active state was much slower and resulted in weaker affinity complexes (Fig. 6). This suggested the early stages of interaction between Gα$_s$ and the conformation of the cytoplasmic receptor binding cavity were important for recognition. The pre-active state appeared as a genuine on-path intermediate of activation, with slow complex formation via this occluded receptor state requiring considerably greater induced fit. In contrast, initial interaction of Gα$_s$ with the cytoplasmic binding cavity of the active state, which was predominantly populated when bound to full agonists, rapidly led to the more expanded ternary complex conformation following TM6 displacement. Although we cannot exclude that ternary complex formation in the presence of partial agonists might proceed via a very low populated active state of the receptor, the observation of G protein complex formation with partial agonist bound β$_1$AR-W implied this was highly unlikely as the active state was expected to be inaccessible under these energetically unfavourable

conditions. Modulation of the pre-active/active state energy barrier between receptors may explain minor occupation of the active state with high efficacy partial agonists for $\beta_2$AR and $A_{2A}$R[11,21,25] and provide a structural basis for allosteric modulator activity, including effects of native lipid environments[9,65,66,70]. Additionally, these differences in both the pre-active/active state equilibrium and ternary complex binding kinetics with partial agonists relative to full agonist could provide a structural and kinetic basis for characterising GPCR therapeutics.

Ternary complex formation with $\beta_1$AR was observed to proceed significantly faster ($k_{on}$) for $G_s$, the canonical binding partner (primary coupling), than for the non-canonical members from the $G_{i/o}$ and $G_q$ families (secondary coupling), leading to an increased binding affinity (Fig. 4). In contrast, the NMR spectra indicated only limited $\beta_1$AR conformational variability between the different mini-G protein ternary complexes (Fig. 5). Whilst this revealed a structural adaptability of $\beta_1$AR that supported functional signal propagation through multiple G proteins, the similarity of the complexes suggested coupling selectivity was not encoded in the final complex conformation (Fig. 6). Cryo-EM structures of $\beta_1$AR reveal a larger interface size with $G\alpha_s$ than $G\alpha_i$[27], but our dissociation kinetics data suggested complex stability was comparable between all ternary complexes with mini-G proteins. This indicated the overall conformation and differing interacting surface and residues between the mini-G proteins provided similar overall enthalpic contributions to the final ternary complex. Based on our data, we suggest that the differential favourability of G protein binding stems from the kinetic advantage in association of the canonical binding partner with the activated receptor. We propose a kinetic model whereby favourable interaction of $G_s$ with the active state results in rapid binding to the receptor, outcompeting the G proteins from other families and resulting in G protein selectivity. The key interactions between G proteins and the $\beta_1$AR active state which contribute to selectivity in this initial interaction are currently unclear and require further study. Changes to binding affinity between mini-$G_s$ and the chimeric mini-$G_{s/i}$ and mini-$G_{s/q}$ proteins indicated only a few mutations to the interacting residues in the $\alpha$5 helix were sufficient to significantly affect binding rates, implicating this region in selectivity in partial agreement with the literature[28,71,72]. Significant conformational changes in the $G\alpha$ protein $\alpha$5 helix are associated with coupling to GPCRs[73] and therefore the G protein conformation in the transient interaction are uncertain. We also noted differences between complexes in the IL4 region, potentially contributing to selectivity through differential interactions with different G proteins. Mutation of the five residues following the NPxxY motif in $\beta_1$AR to the equivalent residues in $\alpha_{2A}$-AR was shown to significantly reduce $G_s$ coupling without substantially affecting $G_i$ coupling[27], thus initial interactions with the IL4 region warrant further investigation.

Although our work emphasises the value of spectroscopic studies in solution for $G_s$ primary coupling, future investigations will establish whether the same kinetic observations also apply to receptors with primary coupling partners from other families. Real-time measurements of ternary complex formation for other GPCRs could help establish general principles of kinetics in G protein selectivity and aid in corroborating selectivity as measured by TRUPATH[46] or similar cellular assays[5,74,75]. Further experimentation to define interactions with the active state could also provide targets for pathway-specific allosteric modulators by stabilising or weakening specific interactions involved in selectivity. Whilst our study was conducted with structurally validated mini-G proteins[47,48] and fully functional receptor with minimal thermostabilisation, additional kinetic studies using full length receptor inclusive of full-length IL3 and C-terminus with heterotrimeric G proteins, including the $\beta\gamma$-subunits which subtly and differentially contribute to interaction interfaces, could aid in revealing further G protein family dependent kinetic differences. Additional receptor states have been suggested for nucleotide-free ($G_0$) as well as

GDP or GTP bound G protein[11,14,76] and thus further work with the full $G\alpha$ subunits could facilitate examination of these states in the $\beta_1$AR activation pathway and in selectivity.

In summary, we showed complex formation of agonist bound $\beta_1$AR with different G proteins was mediated through conformational selection of the active state but required further induced fit to reach similar final ternary complex conformations. $\beta_1$AR showed conformational adaptability that facilitated engagement with partners from the $G_s$, $G_{i/o}$, and $G_q$ families, supporting signal transmission through multiple pathways and satisfying the mechanistic requirements of different G proteins through refinement of final ternary complex conformations. However, favourable interactions with the active state of the agonist-bound $\beta_1$AR greatly accelerated primary coupling over less favourable interactions with secondary couplers, implicating kinetics of complex formation as a major contributing factor to G protein selectivity. Our study emphasises the necessity for investigations into the initial transient interaction between the elusive GPCR active state and G proteins using a range of spectroscopic techniques to extend our understanding into the molecular basis of G protein selectivity.

## Methods
### $\beta_1$AR constructs
Receptor constructs were based on the minimally thermostabilised turkey $\beta_1$AR-Met$\Delta$5 (here, $\beta_1$AR-W) construct used in previous work[20]. The $\beta_1$AR-W construct differed from wild type turkey $\beta_1$AR by truncation of the N-terminus ($\Delta$3-32), C-terminus ($\Delta$368-483) and IL3 ($\Delta$244-271). Single residue changes included C358A$^{8.59}$ to remove a palmitoylation site, C116L$^{3.27}$ to increase expression yield[77], thermostabilisation mutations[37] R68S$^{1.59}$, E130W$^{3.41}$, F327A$^{7.37}$ and F338A$^{7.48}$ and methionine mutations M44L$^{1.35}$, M48L$^{1.39}$, M179L$^{4.63}$, and M281A$^{6.26}$ to reduce spectral crowding[20]. The E130W$^{3.41}$ mutation[39] was reverted to E130$^{3.41}$ in the $\beta_1$AR-E construct. For some experiments the mutation L289M$^{6.34}$ was introduced as further NMR probe, see results for its use. $^{19}$F NMR experiments utilised modified $\beta_1$AR-W and $\beta_1$AR-E for site specific labelling with $^{19}$F probes, as previously demonstrated[19]. Off-target labelling was avoided via mutagenesis of C85V$^{2.48}$ and C163L$^{4.47}$. Experiments with $^{TET}$C344$^{7.54}$ used labelling of the native C344$^{7.54}$ residue and experiments with A282C$^{BTFA,6.27}$ used a construct with additional C344S$^{7.54}$ and A282C$^{6.27}$ mutations. All new mutations were introduced using site directed mutagenesis via PCR with Q5 polymerase (NEB). PCR products were phosphorylated with PNK (Thermo), ligated with ligase (Thermo) and the template was removed via Dpn1 (Thermo) digestion. Ligated PCR products were transformed into E. coli (DH5$\alpha$) and amplified plasmids were purified by PureLink miniprep (Invitrogen). Primers used for mutagenesis in this study are listed in Supplementary Table 8.

### $\beta_1$AR expression and purification
Receptor constructs were expressed and purified in lauryl maltose neopentyl glycol (LMNG) as previously described[19,20]. In brief, Spodoptera frugiperda (Sf9) cells (Thermofisher, 12659017) were transfected with pBacPAK8 plasmids containing the relevant $\beta_1$AR construct and flashBac (OET) expression vector using CellFectin II reagent (Gibco) to generate baculoviral particles. Sequential infections of increasing volume were carried out to amplify baculovirus. Sf9 cells in mid-log phase were diluted to $3 \times 10^6$ mL$^{-1}$ and receptor was expressed by infection with viral stocks at a 1:200 dilution. $^{13}$C-methionine labelled receptor was expressed in ESF 921 $\Delta$ methionine deficient media (Expression Systems) supplemented with penicillin (50 U mL$^{-1}$), streptomycin (50 $\mu$g mL$^{-1}$), and amphotericin B (50 $\mu$g mL$^{-1}$). $^{13}$C-methyl-methionine was added at 0.25 g L$^{-1}$ 4 h post baculoviral infection and cell density was diluted to $1.5 \times 10^6$ mL$^{-1}$. Unlabelled receptor was expressed at $1.5 \times 10^6$ mL$^{-1}$ in Insect-XPRESS with L-glutamine (Lonza) supplemented with 5% heat inactivated FBS (Sigma), and the

above antibiotics. Infected cells were incubated at 27 °C with shaking for 48 h and harvested via centrifugation at 4000 g. Cells were washed once with PBS and centrifuged before storage at −20 °C until receptor purification. All purification steps were performed at 4 °C unless otherwise stated. Cells were thawed at RT and resuspended at 4 °C in solubilisation buffer (20 mM TrisHCl, pH 8.0, 350 mM NaCl, 3 mM imidazole, 1.5% LMNG, and 1x protease inhibitor cocktail (Roche)) for 1 h with constant stirring. Lysed cells were centrifuged at 126,000 g for 45 min. The supernatant was collected and sonicated (Fisherbrand 505) on ice at 25% amplitude for a total of 2 min, with pulse on and off times of 5 s. Clarified lysates were applied at 1 mL min$^{-1}$ to a 5 mL NiNTA column (Cytiva) and washed in 20 mM TrisHCl, pH 8.0, 350 mM NaCl, 50 mM imidazole, and 0.02% LMNG. Receptor was eluted from the column in 20 mM TrisHCl, pH 8.0, 350 mM NaCl, 250 mM imidazole, and 0.02% LMNG. The elution fractions were pooled and concentrated with a 50 kDa MWCO Amicon concentrator. Receptor was applied to a Superdex 200 Increase 10/300 column (Cytiva) in 20 mM TrisHCl, pH 8.0, 150 mM NaCl and 0.02% LMNG and the resulting peak fractions were pooled and concentrated, generating receptor samples of >95% purity.

## $^{19}$F labelling

Receptors were purified by NiNTA affinity chromatography as above, and then buffer exchanged 10-fold into labelling buffer (20 mM TrisHCl, pH 8.0, 350 mM NaCl, and 0.02% LMNG). A282C$^{6.27}$ was labelled with 3-bromo-1,1,1-trifluoroacetone (BTFA) due to the stability of the C-S bond formed therefore reducing hydrolysis at the highly solvent exposed TM6 cytoplasmic tip of β$_1$AR-E. Receptor concentration was adjusted to 20 µM and the sample incubated at 4 °C overnight in the presence of 200 µM BTFA (Sigma) and 50 µM glutathione disulfide (GSSG), then buffer exchanged 10-fold into labelling buffer. C344$^{7.54}$ was labelled with 2,2,2-trifluoroethanethiol (TET) as previously reported[19] as the position was much less prone to hydrolysis. The receptor was buffer exchanged into labelling buffer and concentrated to 10 µM before being incubated at 4 °C with 100 µM Aldrithiol-4™ (Sigma) and 50 µM GSSG for 10–15 min. Buffer exchange was repeated, and the receptor was concentrated to 10 µM and incubated at 4 °C with 100 µM TET and 50 µM GSSG for 10–15 min before a final buffer exchange into labelling buffer. $^{19}$F labelled proteins were concentrated to 1 mL and applied to a Superdex 200 Increase 10/300 column running with 20 mM TrisHCl, pH 8.0, 150 mM NaCl, and 0.02% LMNG. The resulting peak fractions were pooled and concentrated for NMR experiments.

## Mini-G expression and purification

Mini-G protein[48] plasmids used here were mini-G$_s$ 393, mini-G$_{i1}$ R46, mini-G$_{o1}$ R12, and chimeras mini-G$_{s/i}$ R43 and mini-G$_{s/q}$ R70. Constructs were identical to those described in Nehmé et al. aside from an additional C96A mutation made to mini-G$_s$ 393 and mini-G$_{s/i}$ R43 to reduce disulfide mediated dimerisation. Mini-G proteins were expressed in BL21-RIL cells (New England Biolabs). Starter cultures were grown overnight in LB supplemented with ampicillin (50 µg mL$^{-1}$), and used to inoculate 500 mL of LB supplemented with ampicillin in 2 L baffled flasks. Cultures were incubated at 37 °C with shaking at 220 rpm until an OD$_{600}$ of 0.6. Expression was induced with 1 mM IPTG, and the culture was incubated at 25 °C overnight with shaking. Cells were harvested by centrifugation at 4000 g for 15 min. Cell pellets were resuspended in PBS and centrifuged before short term storage at −20 °C. All purification steps were performed at 4 °C unless otherwise stated. Cell pellets were thawed and resuspended into lysis buffer (40 mM TrisHCl, pH 7.4, 100 mM NaCl, 10 mM imidazole, 10% glycerol, 5 mM MgCl$_2$, 50 µM GDP, 1 mM PMSF, 100 µM DDT, 50 µg mL$^{-1}$ lysozyme, and 1 × Roche protease inhibitor cocktail). The cells were lysed with 3 passes through an EmulsiFlex-C5 (Avestin) operating at ~500 bar. Lysates were clarified at 75,000 g for 30 min and the

supernatant was sonicated on ice at 40% amplitude for a total of 2 min, with pulse on and off times of 5 s. The lysate was applied to a 5 mL NiNTA column at 3 mL min$^{-1}$. The column was washed in 40 mM TrisHCl, pH 7.4, 100 mM NaCl, 10 mM imidazole, 10% glycerol, 5 mM MgCl$_2$, and 50 µM GDP, followed by a second wash in 20 mM TrisHCl, pH 7.4, 500 mM NaCl, 40 mM imidazole, 10% glycerol, 1 mM MgCl$_2$, and 50 µM GDP. The mini-G was eluted in 20 mM TrisHCl, pH 7.4, 100 mM NaCl, 500 mM imidazole, 10% glycerol, 1 mM MgCl$_2$, and 50 µM GDP. Peak fractions were pooled, supplemented with 5 mM TCEP and His tagged tobacco etch virus protease (TEV, made in-house) at a 1:30 (mg TEV:mg mini-G) dilution, and dialysed against 2 L of dialysis buffer (20 mM Tris pH 7.4, 100 mM NaCl, 10% glycerol, 1 mM MgCl$_2$, 10 µM GDP) in a 3.5 kDa cutoff dialysis membrane (Spectrum Labs) for 20 h. The sample was supplemented with 20 mM imidazole and passed over a gravity flow column packed with 5 mL of NiNTA resin (QIAGEN). The column was washed with 5 mL of dialysis buffer plus 20 mM imidazole, and the flow through was collected. The protein was then concentrated to 2.5 mL using a 10 kDa MWCO concentrator and applied to a HiPrep 16/60 sephacryl S200 HR column (Cytiva) running in 20 mM TrisHCl, pH 8.0, 150 mM NaCl, 1 mM MgCl$_2$, 1 µM GDP, and 1 mM TCEP. When preparing Mini-G$_s$ 393 and mini-G$_{s/i}$ R43 with the C96A mutation, TCEP was omitted. Peak factions were pooled and concentrated to ~1.5 mM mini-G, except for Mini-G$_{i1}$ which was concentrated to ~680 µM due to poor stability.

## Nb6B9 expression and purification

Procedures to obtain nanobody Nb6B9 were as previously reported[19]. Briefly, expression was carried out in BL21-RIL cells grown in LB supplemented with kanamycin (50 µg mL$^{-1}$). Once cultures achieved an OD$_{600}$ of 0.8 at 37 °C, expression was induced posthaste by addition of 1 mM IPTG and temperature was reduced to 25 °C for 16 h with shaking at 200 rpm, after which cells were harvested by centrifugation. Cells were resuspended in 20 mM TrisHCl, pH 8.0, 150 mM NaCl, and 1 × Roche protease inhibitor cocktail, and lysed with 3 passes through an EmulsiFlex-C5 operating at ~500 bar. The lysate was clarified at 75,000 g for 20 min and sonicated on ice at 40% amplitude for a total of 2 min with pulse on and off times of 5 s. The lysate was passed over a 5 mL NiNTA column and washed with 20 mM TrisHCl, pH 8.0, 150 mM NaCl, then washed with 20 mM TrisHCl, pH 8.0, 150 mM NaCl, and 6 mM imidazole. Nb6B9 was eluted from the column with 20 mM TrisHCl, pH 8.0, 150 mM NaCl, and 250 mM imidazole. The eluted Nb6B9 was dialysed against 2 L of 50 mM sodium acetate, pH 4.8, and 75 mM NaCl, then applied to a Resource-S cation exchange column. Bound protein was eluted with a linear NaCl gradient up to 1 M. Purified Nb6B9 was then buffer exchanged into 20 mM TrisHCl, pH 8.0 and 150 mM NaCl using a desalting column (Cytiva) and concentrated to ~1.2 mM.

## NMR experiments

NMR samples containing $^{13}$C-*methyl*-methionine labelled or $^{19}$F labelled β$_1$AR at 30–60 µM and supplemented with 5% D$_2$O were prepared in 5 mm Shigemi tubes. Receptor was prepared in the apo form or with ligand (isoprenaline, xamoterol, salbutamol) at 1 mM to ensure the ligand-bound receptor population was always >99.9% and changes to chemical shifts and peak intensities reflected only the conformational exchange of the ligand-bound state. 2 mM Na-ascorbate was added to slow down ligand oxidation. Mini-G or Nb6B9 proteins were added at 2 or 8 molar equivalents. In the case of mini-G complexes, 15 mM MgCl$_2$ was added. Spectra of ternary complexes formed using mini-G protein that was preincubated with apyrase (5 units, 2 h) looked very similar to ones without apyrase treatment (Supplementary Fig. 19). Hence, all NMR spectra in this work were recorded without any prior treatment with apyrase, consistent with the BLI experiments. $^1$H-$^{13}$C 2D NMR spectra were performed on a Bruker Avance III 800 spectrometer ($^1$H 800 MHz) running TopSpin 3.1 and equipped with a 5 mm TXI HCN/z

cryoprobe or where specified a Bruker Avance III 950 spectrometer ($^1$H 950 MHz) (The Francis Crick Institute, London) equipped with a 5 mm TCI HCN/z cryoprobe. NMR spectra were recorded at 308 K using the XL-ALSOFAST sequence[78] with experiments typically run with 64 scans (~2 h) using 100 increment pairs in the indirect dimension ($t_{1H}$ 50 ms, $t_{13C}$ 25 ms) and with a spectral width of 10,000 Hz (offset: 3750 Hz) in direct dimension and 4000 Hz (offset: 3420.27 Hz) in the indirect dimension. To improve the signal-to-noise ratio, multiple experiments were co-added. Data were processed in Azara v2.8 (W. Boucher) and analysed using CCPN Analysis V2[79]. $^{19}$F 1D NMR spectra were recorded at 308 K or 298 K where specified using a Bruker Avance III 600 spectrometer running TopSpin 3.1 and equipped with a 5 mm QCI HFCN/z cryoprobe ($^{19}$F 564 MHz), or where stated 700 MHz ($^1$H) Bruker Avance III spectrometer equipped with a 5 mm TCI HCN/z cryoprobe ($^{19}$F 658 MHz), tuneable to $^{19}$F (The Francis Crick Institute, London). $^{19}$F chemical shifts were referenced against an internal standard of 2 µM trifluoroacetic acid (TFA) at −75.22 ppm. Pulse acquire experiments were run recording 2560 complex points (50 ms) with a spectral width of 50,000 Hz (offset: −50,821.80 Hz) and consisting of 5000 to 50,000 scans with a recycling time of 1 s, which lasted between 2 and 12 h to achieve acceptable signal-to-noise. All 1D FIDs were apodized with 20 Hz line broadening and zerofilled to 64k points prior to FFT using Topspin 3.1. All signals were deconvoluted as Lorentzian lines for the measurement of linewidths at half height and the comparison of $R_2$ values. Overlapping signals were deconvoluted using in-house written software. Fitted curves produced residuals that deviated from the baseline no more than the expected noise level. Solvent accessibility of the $^{19}$F probes was assessed by 1D NMR through addition of increasing concentrations (0, 1, 2, 3 mM) of the Gd$^{3+}$ paramagnetic relaxation agent gadopentetic dimeglumine (Magnevist). Signal intensities and $R_2$ values were analysed as a function of Gd$^{3+}$ concentration. $R_2$ values were calculated from $I(t_1) = I(t_2) e^{-R_2(t_2 - t_1)}$, where $I(t_1)/I(t_2)$ was obtained using a two-point relaxation measurement comparing the intensities in a CPMG experiment ($\nu_{CPMG} = 5000$ Hz) with the spin-echo reference experiment, each with spectral widths of 50,000 Hz and ranging from 5000 to 30,000 scans. Full $^{19}$F CPMG relaxation dispersion data were measured in a 1D constant time implementation with a transverse period of 2.5 ms (10 ms) (2 points, $\nu$CPMG 400 (100 Hz) $vs$ 5000 Hz). Exchange between pre-active and active states was investigated using 1D saturation transfer experiments recorded at 564 MHz ($^{19}$F). The observed peak was placed on-resonance, and the saturated peak was irradiated with a weak RF field (saturation field strength 25 Hz, for 1 s) before a 90° hard pulse and acquisition using a spectral width of 10,000 Hz and 5000 scans. A saturation spectrum and a reference spectrum were recorded for each experiment, with saturation pulses placed symmetrically about the offset (see Supplementary Figs. 8 and 13 legends for details). A 2 s recovery delay was used. Pulse-acquire $^{19}$F NMR experiments were recorded before and after saturation transfer experiments to confirm sample integrity.

### In cell ligand affinity measurements

HEK293T cells (ATCC, CRL-3216) were transiently transfected with FLAG-Nluc-β$_1$AR. Cells were grown overnight, harvested and seeded at 10,000 cells per well onto PLL-coated white 96-well plates (Greiner Bio-One). Media was removed after 24 h and cells washed with PBS before addition of 90 µl PBS containing 0.49 mM MgCl$_2$·6H$_2$O, 0.9 mM CaCl$_2$·2H$_2$O, 0.1% BSA and 0.11 µM Nano-Glo® Substrate. The plate was incubated for 10 min at room temperature in the dark. Cells were stimulated with varying concentrations of (S)-propranolol-red (CA200689, Hello Bio), from 0.1 nM to 316 nM, in the absence or presence of 10 µM unlabelled propranolol. Data was collected using BMG Labtech PHERAstar (software Version1.60 R4, Firmware version 1.33). Emission was measured at 460 nm and >610 nm for 30 min, every 40 s and the BRET ratio (Em. $\lambda$ > 610 nm/Em. $\lambda$ 460 nm) corrected to vehicle-treated cells. Specific binding of each concentration of (S)-

propranolol-red was determined by subtracting the vehicle-corrected, ΔBRET values for non-specific binding from total binding at 20 min after addition of (S)-propranolol-red ± propranolol to allow all responses to plateau. Specific binding data were fitted to the one-site specific binding model in GraphPad Prism 9.4 to calculate the $K_D$ for (S)-propranolol-red at each FLAG-Nluc-β$_1$AR variant.

The p$K_i$ for isoprenaline at each FLAG-Nluc-β$_1$AR variant was determined by transfecting cells and seeding as described above. After 10 min incubation of cells in 90 µl PBS containing 0.49 mM MgCl$_2$·6H$_2$O, 0.9 mM CaCl$_2$·2H$_2$O, 0.1% BSA and 0.11 µM Nano-Glo® Substrate, cells were stimulated with varying concentrations of isoprenaline, from 10 nM to 100 µM, in the absence and presence of 2 nM (S)-propranolol-red. Emission was measured at 460 nm and >610 nm for 30 min, every 60 s and the BRET ratio (Em. $\lambda$ > 610 nm/Em. $\lambda$ 460 nm) corrected to vehicle-treated cells. ΔBRET values at 20 min were fitted to the one-site fit $K_i$ model in GraphPad Prism 9.5.1 to determine affinity (p$K_i$) values and are represented as mean of three separate experiments ± SEM.

### TRUPATH assays

HEK293T cells were transiently transfected with β$_1$AR, each Gα-Rluc8, and appropriate Gβ and G$\gamma$[46], at a ratio of 1:1:1:1. For G$_q$, a constitutively active mutated version (R183Q) was used to increase the sensitivity for detection of G$_q$ activation[80]. After 24 h, cells were harvested and seeded onto PLL-coated white 96-well plates (Greiner Bio-One) at a density of 50,000 cells per well. Media was removed after 24 h, and cells washed with HBSS before addition of 90 µl HBSS containing 20 mM HEPES, 0.1% BSA, adjusted to pH 7.4, and 11.1 µM coelenterazine 400a (Nanolight Technology, USA). The plate was then incubated for 10 min at room temperature in the dark. Cells were stimulated with varying concentrations of isoprenaline, from 10 pM to 10 µM, and emission measured at 400 nm and 515 nm for 20 min, every 60 s. Emission measurements was made using BMG Labtech PHERAstar (software Version1.60 R4, Firmware version 1.33). The BRET ratio (Em. $\lambda$ 515 nm/ Em. $\lambda$ 460 nm) was corrected to vehicle-treated cells and the area under the resulting curve used to produce the dose-response curves shown. Dose-response curves were fitted to the 3-parameter logistic equation to determine Emax and pEC$_{50}$ values for isoprenaline at each β$_1$AR variant and are represented as mean of three experiments ±SEM.

### Biolayer interferometry (BLI) measurements

β$_1$AR was immobilised to streptavidin (SA) coated bio-sensor tips (Octet) through an avi-tag introduced at the N-terminus of receptor constructs via mutagenesis. Avi-tagged receptors were purified as described above and diluted to ~40 µM. Site specific labelling of the avi-tag with biotin was performed by 2 h incubation at room temperature with 10 mM ATP, 10 mM Mg acetate, 50 µM biotin and 1 µM GST-BirA (prepared in house). Receptor was then repurified and buffer exchanged via SEC using a Superdex 200 Increase 10/300 column running with 20 mM TrisHCl, pH8.0, 150 mM NaCl, and 0.02% LMNG. BLI binding assays were performed in BLI buffer (20 mM TrisHCl, pH 8.0, 150 mM NaCl, 15 mM MgCl$_2$, 0.02% LMNG, and 0.1% BSA) and assay mixtures were prepared in black 96-well plates with wells filled to 200 µL. All experiments were run on an 8-channel Octet® R8 instrument (Sartorius) using Data Acquisition V 11.0.0.64 (Pall ForteBio LLC), at 25 °C shaking at 400 rpm. SA coated biosensors were pre-hydrated in BLI buffer for a minimum of 10 min before experiments. Baseline signal was measured in BLI buffer for 120 s before the receptor (300 nM) was loaded on the tips with an association time of 600 s followed by dissociation in BLI buffer for 600 s to eliminate non-specific binding. Receptor-loaded tips were transferred to BLI buffer containing agonist (500 µM) for 300 s. Association steps into BLI buffer plus agonist and varying binding partner concentrations were 300–600 s. Dissociation steps back in BLI buffer plus agonist for 300–600 s. All experiments were run in triplicate using $n = 3$ separate tips and a reference channel

with loading steps using un-biotinylated avi-tagged receptor were used to subtract nonspecific binding (see Supplementary Methods). Appropriate concentrations of binding partners to give traces free of artifacts and good signal-to-noise were pre-determined via a BLI based titration measurement. BLI data was analysed on Data Analysis V 11.0.0.4 (Pall ForteBio LLC) using reference channel subtraction and Savitzky–Golay filtering. Kinetic parameters were determined assuming monophasic 1:1 association and dissociation curves with local partial fitting. Statistical significance was calculated with unpaired two-tailed $t$-tests comparing β₁AR-E to β₁AR-W, comparing β₁AR-E with different ligands, and comparing between mini-G proteins with a Bonferroni correction to the significance level. Dose-response experiments for mini-G$_s$ binding to β₁AR-E and β₁AR-W in the presence of 500 μM isoprenaline were recorded using a mini-G$_s$ dilution series from 31 nM to 2000 nM. Responses were used to calculate $K_D$, $k_{on}$, and $k_{off}$ values using a 1:1 global fitting with R$_{max}$ unlinked by sensors and a steady state analysis of a response curve was used to calculate $K_D$ and nominal R$_{max}$ values. Single point measurements were conducted with a mini-G$_s$ concentration established through pre-inspection of a concentration series that identified a BLI trace with a clear 1:1 monophasic behaviour (Supplementary Table 4). The optimised single point measurement approach was validated by comparison of K$_D$, $k_{on}$ and $k_{off}$ values for mini-G$_s$ binding to β₁AR in the presence of isoprenaline to those obtained by the dose-response experiments, which showed similar values (Supplementary Fig. 4c, Supplementary Table 1). The binding of non-canonical mini-G proteins and mini-G$_s$ in the presence of partial agonists was investigated using the validated single point measurement approach, as these lower affinity interactions required higher mini-G concentrations in the micromolar range to obtain kinetic parameters due to small responses at sub-μM concentrations (Supplementary Fig. 3f) and non-specific aggregation on biosensors above 4000 nM. See the Supplementary Methods for further details on BLI controls, single point measurements, and dose-response experiments.

**Reporting summary**

Further information on research design is available in the Nature Portfolio Reporting Summary linked to this article.

## Data availability

The NMR, BLI, *in cell* and Trupath ligand binding data generated in this study have been deposited and can be downloaded from the permanent link: https://figshare.com/projects/Data_sets_for_-_Structurally_similar_G_protein_complexes_with_1-adrenergic_receptor_active_state_show_differential_binding_kinetics_mediating_selectivity_/177996 or by contacting the authors. The BLI, *in cell* and Trupath ligand binding data generated in this study are provided in the Source Data file. The PDB accession codes for the structures used in the interpretation of NMR data in this study are 7JJO, 7SOF, 6KR8, 6H7J, 6BVQ, 3SN6, 6EG8, 2YCY, and 2Y03. Source data are provided with this paper.

## Code availability

The in-house written software (R.W. Broadhurst, T. H. Harman, unpublished) used for deconvolution of $^{19}$F NMR spectra was custom-made for this study and is available upon request by contacting the authors. The NMR processing package Azara v2.8 can be obtained through the link https://cambridge2000.com/azara/ where the software is available under licence directly from Dr W. Boucher.

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

## Acknowledgements
This work was funded through BBSRC research grants to DN (BB/W020718/1) and to GL (BB/W014831/1). GL is a Royal Society Industry Fellow (NF\R2\212001). THH is the recipient of an AstraZeneca studentship (Cambridge, UK). We are grateful for access to the Biophysics facility and the Biomolecular NMR facility of the Department of Biochemistry (U. Cambridge) and the MRC Biomedical NMR Centre of The Francis Crick Institute (London). We are grateful to Dr Chris Tate and Pat Edwards for providing the mini-G constructs. We are grateful to the Hyvönen group, Dept. of Biochemistry, University of Cambridge, for providing the biotinylated avi-tagged SARS-CoV2 Spike protein Receptor Binding Domain for BLI controls.

## Author contributions
AJYJ, THH, GL, DN designed the research. AJYJ, THH, MH, OEL, DN performed research. AJYJ, THH, OEL, MH, DN analysed data. THH, DN, AJYJ wrote the manuscript (all authors edited the manuscript). DN supervised the project.

## Competing interests
The authors declare no competing interests.
