## [Peer Review File · Nature Communications]

Binding kinetics drive G protein subtype selectivity at the β 1-adrenergic receptorREVIEWER COMMENTS

Reviewer #1 (Remarks to the Author):

The authors present a study on the details of the interaction of β 1AR and different miniG proteins. They introduce a new β 1AR construct, β 1AR-E, which is less thermostabilized towards the inactive state, and thus accesses the active receptor state more easily and reveals interesting details on its conformation. Using primarily this construct, the physico-chemical characteristics of miniG binding to the receptor are studied. The main findings are (i) the new construct itself, (ii) the fact that in the active state the large outward motion of TM6 doesn't occurred yet and thus requires a conformational change of the receptor for G-protein binding, and (iii) that the selectivity towards miniGs over other studied miniG variants is dominated by a faster on-rate.

These findings are of high importance for the entire receptor community: (i) the β 1AR-E construct will be a valuable tool for biophysical studies, (ii) the insight into the active state conformation corroborates a similar finding for β 2AR and deepens our understanding of the pathway leading to G-protein engagement. (iii) Finally, the finding that G-protein selectivity seems to be governed by the on-rate is highly exciting and shall spur a number of follow-up studies. The determined k_{on} , k_{off} and KD parameters will be very valuable for modelling and understanding the mechanism of selectivity.

In general the manuscript is well-written and the experiments are carried out in a highly diligent manner (as always from this group). I only felt that I had to read through a lot of detail, where the authors describe their system, before getting to the meat of the paper. I wonder if part of the basic characterization of their system can be moved to the supplementary information and only a short summary is presented in the main manuscript.

Apart from this, I only see minor modifications required before publication (it's rare that I have so few comments to a manuscript).

These points may require further attention:

Which binding partner was used to produce the data shown in figure 1d?

The KD , k_{on} and k_{off} values that were experimentally determined only appear as column diagrams in the main manuscript. The logarithmic scale makes it very difficult to read the exact values. Please include the found values in the figure in the main article. It's difficult to find the corresponding values in the supplement.

The authors make several statements on whether a binding event progresses via an induced fit or a conformational selection mechanism. To me the data is not sufficient to exclude one or the other mechanism, although one might be more likely than the other. For example, the data indicates that the cytoplasmic binding cavity is not fully formed in the active state. The conclusion of the authors is that G-protein binds to the partially formed cavity through conformational selection and then TM6 is driven

outwards via an induced fit mechanism. However, there is no clear indication that TM6 might not visit the outward-oriented conformation for a fraction of the time in the active state, rendering binding a mechanism entirely based on conformational selection.

A direct binding of G-proteins to the pre-active state is suggested, i.e. to β 1AR-W bound to a partial agonist. Couldn't the lowered on-rate equally be interpreted in such a way that binding still proceeds through the active state, since this state has a low population in the β 1AR-W situation?

A typo: In line 251, it should probably say ...peak locations THAT were reminiscent..., or ...peak locations reminiscent...

Reviewer #2 (Remarks to the Author):

This manuscript focuses on GPCRs, which are a significant class of membrane proteins involved in transmitting various signals across cell membranes, leading to diverse downstream signaling pathways affecting physiological processes. GPCRs interact with heterotrimeric G proteins and are known to exhibit selective coupling with specific G protein subtypes. Understanding the mechanisms behind this selectivity is essential for comprehending cellular signaling and developing GPCR-targeted therapeutics. The study specifically focuses on the β 1-adrenergic receptor (β 1AR), which predominantly couples to Gs but also interacts with Gi. The well-structured design utilized a set of complementary biophysical methods, allowing it to shed light on the structure, dynamics, and kinetics of β 1AR. The study employs spectroscopic techniques, including NMR spectroscopy, to investigate the structural dynamics of GPCRs. NMR studies of GPCRs in their active state, coupled to different G proteins, are limited but crucial for a comprehensive understanding of G protein selectivity. It is well-established that agonist binding to GPCRs induces structural changes in the receptor, influencing its activation pathway. However, the exact conformation of the active state is challenging to determine directly due to its inherent instability and dynamic nature. Previous research identified ligand-dependent changes in the equilibrium between inactive and activation-intermediate receptor states.

In this work, the researchers aimed to restore accessibility to the agonist-bound active state of β 1AR by reverting a stabilizing mutation. Using this modified construct, they examined β 1AR activation and G protein selectivity through NMR spectroscopy. The findings indicate that the active state resembles the conformation found in ternary structures but requires additional conformational rearrangements to reach the solution Gs ternary state. Furthermore, subtle conformational differences between ternary complexes with different G proteins suggest receptor adaptability.

In addition to NMR spectroscopy, the researchers employed bio-layer interferometry (BLI) to investigate the *in vitro* binding of G proteins to β 1AR. This complementary approach provided insights into affinity and binding kinetics under conditions akin to NMR experiments. These binding assays confirmed that complex formation with Gs occurs more rapidly, suggesting a kinetic-driven selectivity gate between

canonical and secondary coupling, driven by the favorability of G protein binding to the receptor's active state. There are a couple of weaknesses in the experimental design as described as well as the resulting analysis and interpretation of the data that requires particular attention to strengthen the conclusions of the paper:

(1) Regarding the in vitro biotinylation step, have you conducted an analysis to assess the level of biotinylation? This would provide a more accurate estimate of the extent of non-specific binding of non-biotinylated material during the BLI experiments.

(2) After capturing the receptor with biotin, did you implement a step to block the streptavidin (SA) surface to reduce the binding of the analyte (= G-proteins)? A clarification on the blocking procedure would be helpful to understand how you approached the reduction of non-specific interactions to allow for proper data interpretation

(3) Could you please provide more clarity on the preparation of the reference channel? The experimental description mentions the use of un-biotinylated avi-tagged receptor for the reference channel. However, it appears that a dissociation phase in BLI buffer was employed to eliminate non-specific binding of un-biotinylated avi-tagged receptor. This leaves some confusion about the reference channel's role. Could you explain how it was prepared and its specific purpose in the experiment?

(4) In light of Figure 2e in the supplementary material, where the dissociation phase drops below the zero response level, it seems indicative of deposition on the reference surface. Considering this, do you agree with my assessment that a properly prepared reference surface with an unrelated but biotinylated protein might be necessary to comprehensively assess the specificity of G-protein binding in your experiments?

(5) The experiments appear to have been conducted at a single concentration, which deviates from the standard procedure where dose-response experiments using multiple concentrations are performed. Using multiple and varying concentrations of G-proteins in your experiments would enable the estimation of the maximum binding signal R_{max} , allowing for the determination of stoichiometry and a more accurate assessment of the level of non-specific binding. Could you discuss the rationale for using a single concentration and whether future experiments should include dose-response studies?

(6) In the data analysis, you assumed monophasic association and dissociation curves with local parallel fitting. Could you please elaborate on the basis for this assumption and whether there are other data or analyses that support this choice? It would be helpful to understand the reasoning behind this specific modeling approach and if there is supporting evidence.

(7) How did you evaluate the structural integrity, folding, and functional activity of the analyte (G-protein)? In other words, how did you ascertain the functional concentration of the analyte, or did you solely rely on the nominal protein concentration for your experiments? What problems would you foresee with this approach and how would that impact the analysis?

(8) In Figure 2d of the supplementary material, it's evident that there are notable discrepancies in the

binding levels of the same G-protein to $\beta 1AR-E$ and $\beta 1AR-W$ constructs. These differences raise questions about the capture levels achieved for both constructs and the nominal R_{max} values expected for a 1:1 interaction in your BLI experiments assuming 100% ligand-binding competent material. Could you please provide a detailed explanation of the capture levels for each construct and the nominal R_{max} values? Additionally, could you elaborate on how these values are reflected in the binding isotherm, shedding light on the observed binding behavior? What R_{max} values have been computed as part of the data fitting approach and how do those values relate to the theoretical R_{max} values ?

Addressing these questions should help to enhance the overall depth and clarity of your study's conclusions. I strongly believe that this would not only bolster the accuracy of your experimental data but also offer a more robust foundation for your findings. This, in turn, would strengthen the validity of the conclusions drawn from your well-designed study, ultimately advancing our understanding of the mechanisms underlying G-protein binding to $\beta 1AR$. Such precision and clarity in your analysis would not only benefit the scientific community but also potentially guide further research in this crucial area of study

Reviewer #3 (Remarks to the Author):

Jones et al. present a thought-provoking study hypothesizing that the primary determinant for G protein subtype selectivity at $\beta ₁AR$ is the on-rate (k_{on}) of G protein binding to the full-agonist activated receptor. G_s is the canonical G protein for the $\beta ₁AR$ and thus likely to outcompete non-canonical G proteins in a cellular environment due to its faster on-rate. The authors show that sequence variations of the C-terminal $\alpha 5$ helix of $G\alpha$ are sufficient to modulate on-rates, while the off-rates (k_{off}) of the tested G proteins were not significantly different. The authors further show that G protein coupling to the full agonist bound $\beta ₁AR$ mainly follows a conformational selection mechanism, followed by a somewhat minor induced fit component. G protein coupling to pre-active (e.g. partial-agonist bound) $\beta ₁AR$, however, followed an induced fit mechanism leading to similar global conformations of the receptor as seen in presence of the full-agonist. The global conformation of the receptor was highly similar when in complex with different G protein subtypes with the most substantial differences found in the TM7 and ICL4 regions, hinting at a more productive $G_s:\beta ₁AR$ interaction.

For this study, the authors reverted the thermostabilizing mutation E130W^{3.41} in the in the previously used (pre-active) $\beta ₁AR-W$ construct to generate the more active $\beta ₁AR-E$ variant. Ligand-binding to both variants was characterised using a BRET-based in-cell ligand-binding assay, while G protein coupling was assessed using the TRUPATH assay. Biolayer interferometry (BLI) was employed to measure the affinity of mini-G variants of canonical and non-canonical $G\alpha$ subunits for both $\beta ₁AR-W$ and $\beta ₁AR-E$. Conformational fingerprints of the different $\beta ₁AR-W$ and $\beta ₁AR-E$ complexes (apo, ligand-bound, and ternary-complexes with mini-G's and a G protein mimicking nanobody Nb6B9) were obtained

by ¹³C-methyl methionine and ¹⁹F NMR using endogenous or introduced probes.

This study is logically structured and exceptionally well written. The NMR data presented is of high quality and sufficiently supports the claims made in this study. The combination of ¹³C-methyl methionine and ¹⁹F NMR is highly complementary, allowing the authors to address specific questions either technique alone would not be able to cover. To my knowledge, this is the first time the kinetics of interactions between a GPCR and different G proteins have been studied using BLI while the KD of mini-Gs binding to β 1AR seems somewhat comparable to previously published data (Nehme et al., 2017; Ref 48). The conclusions made seem to be supported by the data and the discussion sufficiently addresses the limitations of the study.

The research presented by Jones et al. is highly relevant and furthers the understanding of G protein subtype selectivity and ternary complex formation. I highly recommend this manuscript for publication.

General comments/questions:

1. I would suggest a title that better reflects the main findings of the study such as “Binding kinetics drive G protein subtype selectivity at the β 1-adrenergic receptor”
2. This study relies a great deal on kinetic data generated using biolayer interferometry (BLI). Raw data (i.e. BLI binding curves) is only shown for mini-Gs, and for a single concentration of the binding partner (2 μ M) in Extended Figures 2e/f. However, these binding curves are not shown for either Nb6B9 or any of the other mini-G proteins analyzed. Given that the BLI results are crucial for this study, I would have expected to find all binding curves in a supplementary Figure. Furthermore, it is unclear if the BLI experiments were carried out with different binding partner concentrations.
3. I understand that Apyrase was added in NMR experiments to “mitigate variations in GDP binding” but BLI experiments suggest it had practically no effect and Apyrase was thus omitted from BLI experiments. Why the discrepancy?

Specific comments & suggestions:

Lines 157 – 159: I fully agree with the authors assessment of M223 adopting a pre-active conformation in β ₁AR-W, compared to the active conformation in β ₁AR-E. It appears that the Iso-bound M223-W resonance almost sits on a linear trajectory between the Apo M223-E and the Iso-bound M223-E resonance, which would further support the assumption that M223-W is a pre-active state. It would be interesting to compare NMR spectra of partial agonist-bound β ₁AR-E to see where M223 would sit with respect to this trajectory.

Lines 166 – 169: A recent publication (Chashmniam et al., 2021; ChemBioChem doi:10.1002/cbic.202000701) hypothesized that neighbour effects contribute more to the methyl ¹³C of Methionines than internal side chain dihedral angles. According to their theory, the upfield chemical shift change of iso-bound M223 in the ¹³C dimension may also be a result of the ^{6.44} shielding effect.

Line 180: I'm not sure if I agree with “reduced conformational variability”. To me, the single (broadened) peak of M178 in the in the iso-bound β ₁AR-E spectrum suggests faster exchange rates compared to slow exchange in the iso-bound β 1AR-W spectrum. What I find interesting, is that the minor M178 peak in the iso-bound β ₁AR-W spectrum (Supp Fig 4a) shifts to the β ₁AR-E position when bound to iso/Nb6B9 (Supp Fig 4b). Furthermore, the iso/Nb6B9 spectrum seems to show a minor M178 peak near the major peak observed in the iso-bound β ₁AR-W spectrum (Supp Fig 4a).

Lines 188 – 195: The authors claim that the more upfield ^1H chemical shift of M153 of iso-bound β_1 AR-E in Fig 2b is indicative of a more active conformation. However, the M153 ^1H chemical shift of Apo β_1 AR-E (Fig 2e) is located further upfield compared to iso-bound β_1 AR-E in Fig 2b. Does this mean that Apo β_1 AR-E is more active than iso-bound β_1 AR-E?
Line 194: I suggest labelling Fig 2d clearly as corresponding to the β_1 AR-E L289M mutant. A label in the figure will make it easier for the reader to rationalize the absence of the L289M resonance in Figs 2b & 2c.

Lines 198 – 200: It may be helpful to mention that agonist-bound X-ray structures of GPCRs often show TM6 in an inactive (or inactive-like conformation). Interestingly, the b36-m23 mutant used for crystallography (Ref 43) did not seem to harbor the E130W mutation, but Warne et al. argued that their thermostabilizing mutations resulted in preferential adoption of the inactive state regardless of the ligand pharmacology.

Line 209: I understand that PDB 2Y03 is a “poor choice” for visualizing the active state as TM6 largely remains in an inactive conformation. It may be helpful to add an active (in complex with Gs) structure to visualize the extent of TM6 twisting/solvent exposure upon activation.

Lines 211 – 214: Why did the authors choose to do CPMG and not CEST experiments if ms-s timescale exchange is consistent with large TM6 helical movements?

Line 215: Could the authors comment on why BTFA was used for TM6 and TET was used for TM7 labelling and/or add a reference to previous testing?

Lines 257 – 259: Do the authors have a means to estimate experimental errors of peak intensities shown in Supp Fig 6a? For example, if one condition was measured in duplicate, an average error can be determined and extrapolated.

Line 265: Reference 59 refers to β_2 AR. Has this also been shown for β_1 AR?

Lines 306 – 308: How confident are the authors with the deconvolution? A residual error analysis (as described by Kim et al., 2013; JACS doi:10.1021/ja404305k) may be helpful.

Line 532: I would be more specific and use the term GPCRs instead of receptors.

Lines 555 – 557: The authors could employ a ^{19}F saturation transfer experiment as previously carried out by Frei et al., 2022; Nat Commun, doi: 10.1038/s41467-020-14526-3) to probe if a low-populated active state is present in the partial agonist-bound β_1 AR-E solution ensemble.

Line 572: A space is missing after “conformation”.

Lines 572 – 575: I suggest a more granular discussion of the authors findings in comparison with reference 27. The cryo-EM structures in reference 27 used heterotrimeric G proteins while the authors used mini-Ga subunits. The simplified system using only the mini-Ga subunits may very well indicate similar koff rates for canonical and non-canonical G proteins, but the cryo-EM structures in reference 27 clearly suggest G protein subtype-specific β_1 AR:G protein interaction interfaces that extend beyond the $\alpha 5$ helix. Hence, off rates for heterotrimeric G proteins (or wild type Ga subunits) may differ from those observed for the mini-G's.

It may be useful to compare the effect of the mini-G proteins used in this study with that of the wild-type heterotrimeric G proteins on ligand binding (like Han et al., 2023 (Ref 35); Extended Figure 10c). Such an experiment could inform on how well the engineered mini-G proteins reflect their wild type counterparts and potentially give the authors more confidence in their claim that G protein on rates are the main drivers of G protein subtype selectivity even if the full heterotrimers were used. This could also be done in the presence and absence of nucleotides (here I'm referring to lines 601 – 607).

Lines 583 – 586: I suggest using “is in partial agreement” since reference 69 also highlights the

importance of other Ga subunit regions in G protein subtype specificity.

Line 586: Insert "is" between selectivity and in.

Line 591: I suggest writing "was shown to significantly reduce" instead of "was significantly shown to reduce".

Line 622: I suggest to either delete "mildly" or change to "minimally".

Lines 728 – 729: The authors state that 2 molar equivalents of mini-G were used. The Figure legend of Figure 3c, however, states that 8 equivalents of mini-Gs were used in combination with xamoterol and the Figure legend of Figure 5d states that 8 equivalents mini-Gi1 were used. Furthermore, I would recommend using "molar equivalents" instead of just "equivalents" in the Figure legends.

Lines 806 & 807: The authors mention "varying binding partner concentrations". However, Extended Figure 2E only shows representative data using 2 μ M mini-G. Were all BLI experiments carried out at a single mini-G concentration of 2 μ M?

Figure 1d: Please state that the BLI binding traces show mini-Gs binding to the b1AR constructs.

Figure 3: The legend states that for c) 8 equivalents of mini-Gs were used in combination with xamoterol while in f-h) 2 equivalents were used in combination with all ligands including xamoterol. Could the authors explain why different molar equivalents of G protein were used in combination with the same ligand?

Figures 3b and 4: What do the error bars describe, SD or SEM? How many biological (n) and technical replicates were collected?

Supplementary Figure 2F: The K_{D} in the presence of apyrase is given as 6.62×10^{-9} M. However, k_{off} (6.52×10^{-3} s⁻¹)/ k_{on} (9.86×10^4 M⁻¹s⁻¹) = 6.62×10^{-8} M.

Supplementary Figure 9: Are those biological (n) or technical triplicates?

Reviewer #1

We would like to thank the reviewer for reviewing our manuscript and appreciate their very favourable response and strong endorsement of our manuscript and research.

1) The reviewer indicates that there is a lot of detail on system characterization to read through before getting to the 'meat of the paper'

We thank the reviewer for this comment. The level of detail we included to describe our system is an important aspect that we debated and considered extensively during the writing of the manuscript. As Nature Communications draws readership from a wide range of fields with different interests and backgrounds, we decided to portray a more comprehensive description of the background and setting of our work. We feel that, in its current form, the earlier parts of our manuscript provide an appropriate and well-balanced introduction to our work that is suitable for GPCR specialists as well as non-specialist readers from other biological communities. We would be keen to keep this part of the manuscript as is.

2) Which binding partner was used to produce the data shown in Figure 1d?

We apologize for the omission of this information. Figure 1d refers to binding curves of mini-G_s which we are now saying in the figure legend. We have also added a mini-G_s at the top of Figure 1d.

3) In the main manuscript the logarithmic scale makes it difficult to read the BLI-determined K_D , k_{on} , k_{off} values.

To make the data in Fig. 3b and Fig 4c more accessible we have written the values at the top of the bar charts. As before, the values including their standard deviations are given in Supplementary Tables 3 and 5.

4) The reviewer is concerned that the available data is not sufficient to describe the binding events leading to ternary complex formation as conformational selection or induced fit. The reviewer illustrates this with the example of the active state of the receptor, which hypothetically and for a very short time might adopt an open state that would enable G protein engagement through a conformational selection mechanism without any need of induced fit (as proposed by us).

We thank the reviewer for raising this issue. We entirely agree with the reviewer and expect this to remain a difficult and debatable topic for the field for a long time. A brief review of GPCR literature reveals an already substantial number of contradicting statements on conformational selection and induced fit and it is not our intention to add to this controversy further. We agree with the reviewer that in a purely mechanistic sense, without the availability of data on short-lived intermediates (which is beyond the scope of this work) it is impossible to exclude that the active state might briefly change into a fully G protein-compatible conformation that is suited for binding. Indeed, the latter would result in a conformational selection process without any further need of induced fit as the reviewer points out.

We would like to emphasize that throughout our manuscript we adhered to an interpretation of our experimental data in a most direct way based on observable states and we abstained from speculation by referring to hypothetical states that were not directly observable due to their low population or their very short lifetimes. Accordingly, our description relates to a main conformation of the full agonist-bound active state as shown in our spectra. Resulting from this approach, the use of the terminologies induced fit and conformational selection in the context of ternary complex formation accommodates the possibility to differentiate between the sizeable conformational differences observed when involving a partial agonist-stimulated receptor and the minor conformational changes seen with a full agonist-stimulated receptor. In the manuscript we emphasize that the full agonist-stimulated receptor conformation is already much closer to the final ternary complex conformation, requiring less structural

rearrangement than a partial agonist stimulated β_1 AR. Following this rationale, based on the states that we experimentally observe, a full agonist-bound receptor G protein engagement would be more in agreement with a conformational selection model followed by small amounts of induced fit to provide the final adjustments. In contrast the partial agonist-stimulated receptor necessitates substantial conformational adjustments upon formation of a ternary complex, which agrees better with induced fit.

In support of our interpretation, earlier work by the Kobilka group presented the structure of full agonist-bound β_2 AR in complex with a G protein $\alpha 5$ peptide, which showed the receptor in a conformation that was slightly distinct from the one in its final ternary complex form with full G protein (for example Liu et al., Cell 177, 1243-1251 (2019)). The structure which differed also in the way the $\alpha 5$ helix interacted within the β_2 AR binding cavity was presented by the authors as an early on-path receptor intermediate possibly involved in G protein selectivity that would then convert into the final ternary complex conformation. The observation of such an 'early' complex that differs from the final ternary form would suggest that following initial G protein engagement further conformational rearrangements of the active state are required, in agreement with the residual induced fit proposed by us.

We understand that these issues are not completely resolved yet and that's why in the manuscript discussion on page 13 we mention already that a better understanding of the coupling mechanism and also its molecular implications for G protein selectivity will require further in-depth studies of the early stages of receptor G protein interaction.

5) The reviewer suggests that a reduction in on-rate might be the result of a reduced population of the active state rather than binding to the pre-active state.

We agree with the reviewer that this possibility can't be fully excluded, and therefore we already mention this alternative interpretation in our manuscript discussion on page 12/13.

However, following the rationale in our manuscript, G protein engagement by the pre-active state of the receptor seems particularly likely for the partial agonist-bound β_1 AR-W, which as the result of the strongly thermostabilizing StaR-mutations features a receptor that is strongly biased towards the inactive state(s). We find it difficult to envisage therefore that under these conditions β_1 AR-W would have a sufficient population of the active state. Assuming an involvement of the active state we would anticipate its population to be substantially larger for β_1 AR-E so that one would have expected to observe an increase in the k_{on} rate from β_1 AR-W to β_1 AR-E. However, for the partial agonists xamoterol (low efficacy) and salbutamol (high efficacy) we observe almost identical k_{on} rates for the two constructs (Supplementary Table 3). We concluded from this in our manuscript that for the two β_1 AR constructs stimulated by partial agonists the active state is most likely not accessible and initial binding of the G protein takes place with the pre-active state of the receptor, providing the main explanation for the rate reduction compared to full-agonist stimulated receptor.

We further carried out saturation transfer experiments to explore the possibility of a low population of active state in partial agonist-bound receptor populations by saturation of the active state signal position in β_1 AR-E A282C^{BTF_A,6.27} in the presence of xamoterol and observing the effect on the pre-active state signal. There was negligible change to the pre-active state signal (Supplementary Fig. 13a), and from this we concluded that the active state is not occupied under these conditions. Although it is possible that the population is sufficiently low to not be observed, or the exchange rate between active and pre-active states is too slow to be detected in this experiment, we believe these explanations to be unlikely. We also conducted saturation transfer experiments which demonstrated exchange between active and pre-active states for isoprenaline-bound receptor (Supplementary Fig. 8), indicating exchange is observable by this technique. Consequently, we conclude that the active state is not occupied in partial agonist bound receptor.

6) Typo in line 251

Thank you, we have now added 'that'.

Reviewer #2

We would like to thank the reviewer for reviewing our manuscript and providing detailed suggestions to improve our study.

1) Regarding the *in vitro* biotinylation step, have you conducted an analysis to assess the level of biotinylation? This would provide a more accurate estimate of the extent of non-specific binding of non-biotinylated material during the BLI experiments.

We thank the reviewer for this comment and recognise the importance of this issue. We have not assessed the level of biotinylation of the receptor constructs, but we assessed the extent of non-specific binding of material through several controls. We apologise for not including these controls in our original manuscript, and now show these controls in Supplementary Fig. 3. We assessed the extent of non-specific binding of non-biotinylated material through loading of an unbiotinylated avi-tagged receptor and an unbiotinylated β_1 AR-E construct used in the ^1H - ^{13}C NMR experiments, i.e. with no N-terminal avi-tag. With respect to the former, the avi-tagged receptor samples were split into two aliquots prior to the biotinylation step in the purification protocol. One aliquot was treated with the biotinylation protocol and further purified by SEC, then used for the experimental channels in BLI experiments, whilst the other was not treated and only further purified by SEC. This provided a control of unbiotinylated avi-tagged receptor to compare to the biotinylated material. Loading of the biotinylated vs unbiotinylated material, which is now shown in Supplementary Fig. 3b, demonstrated that the unbiotinylated sample showed minimal loading relative to the biotinylated protein, indicating the non-specific binding of non-biotinylated receptor was negligible. The unbiotinylated β_1 AR-E construct lacking the avi-tag also showed negligible loading onto biosensor tips, further supporting the negligible extent of non-specific binding of non-biotinylated material.

We also assessed if biotinylation of the receptor may occur at other Lys residues other than the Lys in the avi-tag. The β_1 AR-E NMR construct without an avi-tag was treated with the same biotinylation protocol as avi-tagged β_1 AR-E. Minimal loading of this sample was observed, shown in Supplementary Fig. 3c, indicating both that there was minimal biotinylation at other sites in the receptor and that non-specific interactions between the biosensor and the receptor were negligible.

2) After capturing the receptor with biotin, did you implement a step to block the streptavidin (SA) surface to reduce the binding of the analyte (= G-proteins)? A clarification on the blocking procedure would be helpful to understand how you approached the reduction of non-specific interactions to allow for proper data interpretation.

We thank the reviewer for bringing this to our attention and apologise for not being clear with our BLI protocols. We have now included a scheme to demonstrate our assay procedure and hopefully clarify how non-specific interactions are reduced. Prior to the loading steps, biosensor tips are incubated in buffer containing 0.1% BSA to block non-specific interactions between receptors and biosensor tips. Following specific loading of receptor and dissociation of any weakly bound material, which is minimal as shown in Supplementary Fig. 3b, the biosensor tips are incubated in buffer containing 0.1% BSA and agonist in a mock association step to act as a blocking step to prevent non-specific association of binding partners (analytes) to the biosensor and further ensure that agonists do not affect the response. Additionally, reference channels with the same analyte conditions as the experimental channels demonstrated that analytes did not interact non-specifically with the biosensors, as shown in Supplementary Fig. 3b.

3) Could you please provide more clarity on the preparation of the reference channel? The experimental description mentions the use of un-biotinylated avi-tagged receptor for the reference channel. However, it appears that a dissociation phase in BLI buffer was employed to eliminate non-specific binding of un-

biotinylated avi-tagged receptor. This leaves some confusion about the reference channel's role. Could you explain how it was prepared and its specific purpose in the experiment?

We apologise for the ambiguity in the role of the reference channel and can clarify this point. The reference channel wells contained exactly the same conditions as the experimental channels with respect to buffer conditions, receptor agonists, binding partner analytes (mini-G), and receptors, aside from the avi-tagged receptor being in the unbiotinylated form which was prepared as detailed in point 1 above. Thus, the reviewer is correct in saying the dissociation phase in BLI buffer eliminated non-specific binding of un-biotinylated avi-tagged receptor and the reference channel confirmed loading of receptor was biotin-specific. Additionally, the reference channel provided a means of examining non-specific association of analytes to the biosensors. This was generally minimal, as shown in Supplementary Fig. 3e and we used the reference channel to subtract this non-specific interaction from the experimental channels. We hope this makes clear the role of the reference channel and we have modified the relevant methods section and have included a Supplementary Methods section on BLI.

4) In light of Figure 2e in the supplementary material, where the dissociation phase drops below the zero-response level, it seems indicative of deposition on the reference surface. Considering this, do you agree with my assessment that a properly prepared reference surface with an unrelated but biotinylated protein might be necessary to comprehensively assess the specificity of G-protein binding in your experiments?

We thank the reviewer for highlighting this point and their suggestion. In our analysis of our experiments, we observed minimal signal from the reference channels, and in rare cases when signal was observed (indicating the deposition suggested), this data was rejected and not used for analysis or interpretation. We did note that deposition on the reference surface was observable at 8 μM of analyte, thus we did not use data above 4 μM analyte (Supplementary Fig. 3e, Supplementary Table 4).

We recognise that this drift may be a confounding factor, and thus we have now repeated this experiment to obtain data without this drift present, which maintains the same experimental observation and conclusion with respect to the absence of effect of apyrase on binding kinetics (Supplementary Fig. 2f). We have further added Supplementary Fig. 17 to show the representative traces for each mini-G and Nb6B9 binding partner.

We also thank the reviewer for their suggestion. We have now performed a control experiment using an avi-tagged and biotinylated construct of the SARS-CoV-2 Spike RBD protein, provided by the Hyvönen group, Dept. of Biochemistry, University of Cambridge, for which we are very grateful. This control experiment demonstrated negligible interaction with the mini-G_s protein (Supplementary Fig. 3d). We suggest these data, alongside the minimal interaction in the absence of agonist ($\beta_1\text{AR}$ in its apo form) (Supplementary Fig. 2e), are sufficient to support this interaction as specific.

5) The experiments appear to have been conducted at a single concentration, which deviates from the standard procedure where dose-response experiments using multiple concentrations are performed. Using multiple and varying concentrations of G-proteins in your experiments would enable the estimation of the maximum binding signal R_{max} , allowing for the determination of stoichiometry and a more accurate assessment of the level of non-specific binding. Could you discuss the rationale for using a single concentration and whether future experiments should include dose-response studies?

We are happy to clarify our rationale for using the single concentration in our BLI experiments. We observed the presence of considerably larger residuals in the k_{on} rates when using higher concentrations of analyte for the higher affinity interaction between the $\beta_1\text{AR}$ -E construct and mini-G_s in the presence of full agonist isoprenaline (Supplementary Fig 4a, b). Interestingly, this behaviour matched well to biphasic behaviour. In contrast, the k_{off} rates were not affected by concentration. As there was only a minor

interaction observed in the absence of ligand (apo, Supplementary Fig. 2e), this demonstrates the binding is specific and functional.

Although it is possible that the mini-G_s is binding with a stoichiometry greater than 1:1 as might be indicated by the biphasic behaviour, it seems likely that the higher affinity interaction is the functional response. Additionally, it is well established in the literature that both G proteins and mini-G proteins associate with receptors in a 1:1 stoichiometry, thus we have no reason to doubt this assumption.

Consequently, we decided to use the single point measurements with concentrations which gave the best match to the monophasic fits with the lowest residuals in all experiments. This was conducted using serial dilution titrations of analyte, generally utilising 3 concentrations (e.g. mini-G_s concentrations of 62.5 nM, 125 nM, 250 nM).

We are thankful for the reviewer's suggestion for using dose-response studies, which we have now provided for the β_1 AR-E and β_1 AR-W binding to mini-G_s and isoprenaline (Supplementary Fig. 4). A concentration series was used as per the standard procedure, and data was analysed using a 1:1 global fit with R_{max} unlinked by sensors. The obtained results for K_D , k_{on} and k_{off} are very similar to the ones of the single-point measurement conducted at an analyte concentration that provided monophasic data (Supplementary Table 1).

However, the much lower affinity interactions between other mini-G proteins and receptor needed substantially higher concentrations as the low nM concentrations used for mini-G_s were insufficient to obtain binding isotherms for low affinity binding partners which required μ M concentrations (see Supplementary Figure 3f).

We found that for these lower affinity interactions the higher concentrations needed to obtain binding isotherms prevented accurate dose-response studies using these conditions. For the much weaker binding of non-canonical mini-G proteins, recording dose-response experiments was problematic, as the elevated analyte concentrations required resulted in data that was qualitatively highly deficient due to non-specific aggregation with the sensor (example shown in Supplementary Fig. 3e). Above 4 μ M, non-specific interactions between mini-G and the biosensor became an issue, which seemed likely to be a deposition. Therefore, as generally the isotherm intensity was too low below 1 μ M of analyte, we decided the single point approach was more applicable for the lower affinity interactions and gave robust and reproducible responses if conducted correctly (see below).

Consequently, we validated the use of a single-point measurement conducted at a pre-selected, 'optimised', concentration using the mini-G_s data. We compared the single point measurements that produced a highly 1:1 monophasic binding isotherm (Supplementary Fig. 17, top row) against the results of a full dose-response analysis (Supplementary Fig. 4) using a 1:1 global fit, and additionally K_D values measured through a steady state analysis of the responses. The comparison of the procedures showed that the values for K_D , k_{on} and k_{off} were similar and generally within one standard deviation (Supplementary Fig. 4c, Supplementary Table 1). We therefore concluded that a binding comparison of the different mini-G proteins based on single point measurements conducted at individually 'optimised' mini-G protein concentrations (as established by pre-inspection of a concentration series that identified a BLI trace with a clear 1:1 monophasic behaviour, Supplementary Table 4) was a suitable experimental procedure, which supported analysis of interactions encompassing the wide range of binding affinities (see Methods and Supplementary Methods). Following this protocol, we obtained highly reproducible results (Supplementary Tables 2,3,5) that enabled a binding comparison for the different mini-G proteins investigated based on the same methodology. Importantly, we chose the single point measurement approach so as to use the same methodology for the high (mini-G_s) and low affinity (non-canonical mini-G) interactions.

In the context of our work, we would like to highlight that small deviations in the absolute values of binding affinities does not alter the main outcome of our manuscript as the numerical differences in k_{on}

and K_D values between the canonical coupling of mini-G_s and the non-canonical coupling of other mini-G proteins differ by several orders of magnitude (Supplementary Table 5).

6) In the data analysis, you assumed monophasic association and dissociation curves with local parallel fitting. Could you please elaborate on the basis for this assumption and whether there are other data or analyses that support this choice? It would be helpful to understand the reasoning behind this specific modeling approach and if there is supporting evidence.

The stoichiometry of the interaction is expected to be 1:1 – numerous structural studies have been performed on many GPCRs and with different G proteins and also mini-G proteins, and these studies have not reported binding to non-canonical binding sites or to multiple G proteins. Therefore, based on the extensive literature, we made the assumption of monophasic association and dissociation curves. As we have mentioned above, we did observe biphasic behaviour at higher concentrations of mini-G_s with isoprenaline and β_1 AR-E, but we hypothesise this could be due to the multiple conformations of receptor with different K_D values for the mini-G_s or simply an artifact, rather than multiple binding sites on the receptor. The local parallel fitting was used as these gave the lowest residuals during data analysis. The more standard global fit was used in the added dose-response studies, which again gave similar K_D values to the single point measurements.

7) How did you evaluate the structural integrity, folding, and functional activity of the analyte (G-protein)? In other words, how did you ascertain the functional concentration of the analyte, or did you solely rely on the nominal protein concentration for your experiments? What problems would you foresee with this approach and how would that impact the analysis?

We thank the reviewer for bringing this important point to attention. The mini-G constructs have all been validated by the original authors (Carpenter, B., & Tate, C. G. (2016). *Protein Engineering, Design & Selection*, 29, 583–593. Nehmé, R., Carpenter, B., Singhal, A., Strege, A., Edwards, P. C., White, C. F., Du, H., Grisshammer, R., & Tate, C. G. (2017). *PLoS ONE*, 12, e0175642.) but we further validated the structural integrity and folding of the mini-G proteins by NMR spectroscopy.

Our ¹⁵N HSQC NMR experiments recorded with different concentrations of ¹⁵N labelled mini-G proteins showed no concentration-dependent variation in linewidth (data not shown). This indicated that there was no evidence of oligomerisation for any of the mini-G proteins used and that based on the estimated rotational correlation time all mini-G proteins were monomeric, in agreement with the literature. The mini-G proteins are stable over a significantly longer period of time than used in the BLI experiments at the same temperature, which we determined by minimal changes to 2D ¹H-¹⁵N TROSY or ¹H 1D experiments. The ¹H NMR experiments show several well resolved peaks for each of the mini-G proteins which can be used to assess the rate of mini-G degradation and structural integrity, in addition to the general fingerprint of the ¹H spectra. An example of a comparison of mini-G_s ¹H 1D experiments is shown in Supplementary Fig. 3g, and demonstrates protein stability at 298 K over 12 h, which is several-fold greater than the time lengths required for BLI experiments. Based on these data, we don't anticipate there to be any issues with the functionality of the proteins used in the BLI experiments, and therefore we felt comfortable in relying on the nominal protein concentrations to reflect the functional concentrations of the analyte.

8) In Figure 2d (2e?) of the supplementary material, it's evident that there are notable discrepancies in the binding levels of the same G-protein to β_1 AR-E and β_1 AR-W constructs. These differences raise questions about the capture levels achieved for both constructs and the nominal R_{max} values expected for a 1:1 interaction in your BLI experiments assuming 100% ligand-binding competent material. Could you please provide a detailed explanation of the capture levels for each construct and the nominal R_{max} values? Additionally, could you elaborate on how these values are reflected in the binding isotherm, shedding light on the observed binding behavior? What R_{max} values have been computed as part of the data fitting approach and how do those values relate to the theoretical R_{max} values?

We thank the reviewer for their comment, and we have now provided data to clarify these issues. The discrepancy in the binding levels between the two receptor constructs likely comes from differences in affinity for the mini-G, and thus differences in the R_{eq} values. The dose response measurements supported calculation of R_{max} values to better analyse differences between the two receptor constructs.

The β_1AR-E and β_1AR-W constructs were captured to similar levels as shown in the added Supplementary Fig. 3b, although β_1AR-W did show subtly reduced capture levels (~10%) relative to β_1AR-E . The R_{max} values calculated from dose response experiments were 1.103 for β_1AR-E and 0.995 for β_1AR-W , and thus a similar difference to the capture level difference of ~10%. Consequently, we suggest that the R_{max} values measured by BLI between the two constructs are similar.

We calculated theoretical R_{max} values by assuming a direct proportionality between the ratio of the molecular weights of analyte to ligand with the ratio of the analyte loading signal to the analyte theoretical R_{max} value. This was based on a similar relationship commonly used in the analysis of SPR experimental data.

The mass of the avi-tagged receptor is approximately 83 kDa, assuming 45 LMNG (~1000 Da each) per receptor micelle (this was calculated based on a molecular weight of receptor-micelle particles determined using translational diffusion NMR experiments, unpublished) and avi- β_1AR to be ~38 kDa. The mini-G_s construct has a mass of ~26 kDa and therefore, with loading of approximately 8.0 for β_1AR-E (Supplementary Fig. 3b), the theoretical R_{max} is ~2.5. Evidently, this is above the measured values for both constructs.

Given the NMR spectra for both β_1AR-E and β_1AR-W constructs bound to isoprenaline and mini-G_s show the receptor is close to fully bound (i.e. in ternary complex) with minimal populations of receptor which don't interact with mini-G, we would expect the BLI experiment to reflect this with respect to functional receptor, and thus the differences in R_{max} are not related to non-functional receptor or to receptor capture levels.

We feel we can justify why there may be differences between the nominal and theoretical values for the R_{max} values.

Firstly, the number of LMNG molecules per receptor micelle on SA biosensors may differ from the value calculated in NMR conditions. A greater number of LMNG molecules per micelle would result in a lower theoretical R_{max} , closer to the experimental value.

Secondly, the assumption of direct proportionality is likely incorrect for this system. The BLI signal intensity is dependent on the interference pattern between the reflection of signal from the protein (biolayer) with a reference surface. Consequently, differences in the sizes and shapes of the proteins may influence this reflection as these factors affect the thickness of the bilayer. In a molecular dynamics simulation of a GPCR in an LMNG micelle (Lee, S. et al. *Biochemistry*. 59, 2125–2134 (2020).), data showed that the LMNG micelle resulted in a deviation from a spherical micelle, resulting in a flattened anisotropic shape. Therefore, the change in the thickness of the bilayer upon β_1AR loading may differ from the assumption of a spherical protein.

Furthermore, in other studies examining the BLI signal from immobilisation of biotinylated DNA aptamers to SA biosensors showed that biotinylating at different sites resulted in differing loading intensities despite the constant molecular weight of the aptamer (Stanborough, T., et al. *ACS Omega*. 6, 6404–6413 (2021)). Consequently, it seems likely that the assumption of direct proportionality is invalid and calculation of theoretical R_{max} values for BLI is more complicated than for similar techniques, such as in SPR analyses.

Thirdly, upon binding of receptor to G protein, the $\alpha 5$ helix inserts into the receptor in the intracellular binding site. Therefore, the change in thickness of the bilayer would be less than anticipated for binding of the mini-G to a surface, resulting in a smaller experimental R_{max} .

Given these challenges, the genuine theoretical R_{max} would be difficult to calculate, and thus we cannot reliably comment on how the experimental R_{max} value compares.

Reviewer #3

We thank the reviewer for the thorough review of our manuscript and their strong endorsement of our work. We are highly appreciative of their comments and suggestions made.

1) Suggestion for a title that better reflects the main findings of the study.

We are grateful to the reviewer for suggesting the pragmatic title. We do like the suggested title and have adopted this as the revised title of our manuscript ('Binding kinetics drive G protein subtype selectivity at the β_1 -adrenergic receptor').

2) Raw BLI data is only shown for mini-G_s and for a single concentration of the binding partner (Supplementary Fig. 2e/f) but not for Nb6B9 or any of the other mini-G proteins.

This comment is similar to an issue raised by reviewer #2. For completeness we have now included BLI binding traces for all the mini-G proteins and nanobody Nb6B9 as shown in the additional Supplementary Fig. 17. We clearly state the conditions used e.g. concentrations of binding partners (Supplementary Table 4) and note that agonists were used at saturating concentrations of 500 μ M.

Each of the binding traces shown represent a measurement at an 'optimised' (see below) single mini-G or nanobody concentration. These single-concentration data were used to obtain k_{on} and k_{off} values from a 1:1 monophasic fit. The k_{on} and k_{off} values from the fits are given in Supplementary Tables 2,3, and 5, and are also mentioned throughout the manuscript and indicated above the bar charts in Figures 3b and 4c.

We are aware that the use of single-concentration data to determine k_{on} and k_{off} is a departure from typical BLI protocols, which usually rely on a global analysis of data obtained at different binding partner concentrations. We justify the use of single-concentration measurements as follows:

In our quantitative BLI comparison using different binding partners, we faced several challenges which required addressing. Mostly, these were related to the large differences in binding affinity (1-2 orders of magnitude) of the complexes formed with the different mini-G proteins. For the various complexes this required the use of very different mini-G concentrations for the BLI measurements (see below). For the isotherms recorded with higher concentrations of analyte during a concentration series with mini-G_s we noticed increasingly biphasic behaviour at higher concentrations of mini-G_s (nM) (see Supplementary Fig. 4).

Although we hypothesised this may arise from at least two different conformations of receptor with different binding affinities for the mini-G_s, i.e. the active and pre-active states, we could not confirm this hypothesis and such behaviour may be an artifact.

Generally speaking, the tighter binding mini-G_s complexes allowed measurements at lower mini-G concentrations. A single-point concentration chosen around 50-100 nM reflected conditions where biphasic behaviour was not a contributing factor (Supplementary Fig. 4).

Complex formation with the other mini-G proteins showed considerably weaker binding (by 1-2 orders of magnitude) and thus to obtain reliable data we had to extend our measurements to much higher concentrations. Isotherms with an appreciable signal were only possible above 1 μ M, but at 8 μ M and above there was non-specific interactions and deposition of the mini-G proteins on the biosensor tips which we did not include in the analysis. Instead we determined 4 μ M as an upper threshold concentration where fits still evidenced clear monophasic behaviour and where the amount of signal readout was sufficient for a reliable fit (in view of the weaker binding). We conducted measurements at several concentrations below this threshold concentration in the hope of a global analysis of this dose response data. However, we realized that the signal intensity (in view of the weaker binding) in that concentration range, i.e. below 1 μ M, was clearly not sufficient to generate reliable multi-concentration data to be included in a global analysis. Instead, the data of the single-point measurement conducted at

the estimated threshold concentration showed to be highly reproducible. Therefore, for the lower-affinity complexes formed with non-canonical mini-G proteins, the concentrations chosen for the single-point measurement were around 1-4 μM and represented a suitable compromise between obtaining sufficient signal and the effects of non-specific interaction with the tips.

In summary, to minimise these issues in view of obtaining a reliable comparison of the different mini-G protein complexes, we decided to go for high-quality monophasic data that resulted from the measurement at a single 'optimised' concentration. Accordingly, for each of the agonist/mini-G conditions investigated, a suitable single-point mini-G binding partner concentration had to be established based on pre-inspection of initial binding curves recorded at 2-3 different concentrations (as typically recorded for dose response measurements). This 'optimised' concentration was established from the curve that provided the smallest fitting residuals to a 1:1 monophasic analysis. A suitable concentration was established for any of the combinations (mini-G/agonist) used (Supplementary Table 4).

In our experience this resulted in highly reproducible and robust data which allowed a consistent comparison of the different mini-G and nanobody complexes. Given we have confirmed physiological coupling to all G protein subtypes with the TRUPATH assays as well as ternary complex formation via NMR experiments, we are confident that, although different concentrations were required to observe coupling of the different G proteins, these couplings replicate the physiological interactions.

While the exact reasons for biphasic behaviour are not entirely clear we have no reason to doubt our monophasic assumptions based on a 1:1 interaction of the proteins forming the complex. In contrast, non-specific interaction of proteins at higher concentrations with the biosensor tips are well-known and should be avoided by conducting the BLI measurements at lower concentration.

We have now also conducted dose-response experiments for mini-G_s with isoprenaline for $\beta_1\text{AR-E}$ and $\beta_1\text{AR-W}$, now described in the Methods and Supplementary Methods, and shown in Supplementary Fig. 4a, b. The data was analysed using a 1:1 global fitting which supported calculation of K_D , k_{on} , and k_{off} values, and was also analysed using a steady state analysis and response curve to calculate R_{max} and K_D values. When we compared the results of the single-concentration measurements with the global fit analysis of a 'classic' dose response measurement, we found the K_D , k_{on} and k_{off} to be highly similar (Supplementary Fig. 4c). Similarly, the steady state analysis of the dose-response experiments gave similar K_D values (Supplementary Fig. 4c).

Having successfully validated for mini-G_s our single-point methodology against the dose-response series as a gold standard, we concluded that the data measured at a concentration that generated monophasic BLI traces would provide the most robust data set for a comparison of the different mini-G binders which varied across a range of affinities by 1-2 orders in magnitude.

Our findings reveal substantial differences in the magnitude of the k_{on} and K_D values between mini-G_s and the other non-canonical mini-G proteins. We are certain that the same trends would be reproduced if for the weaker binding mini-G proteins full dose-response measurements could be conducted coupled to a global analysis. Even if the absolute values might possibly change, the large differences and trends that we observe would be maintained.

Next to the supplementary data mentioned above, we have now also added a Supplementary Methods Section where we are explaining our reasoning behind the single-concentration measurements and where we provide all the detailed measurement conditions and explain how we obtained the data.

3) Apyrase was added in NMR experiments to 'mitigate variations in GDP binding' but BLI experiments suggest it had practically no effect and apyrase was thus omitted from BLI experiments. Why the discrepancy?

We thank the reviewer for bringing this up. There is no discrepancy. However, the current formulation used by us in the description of the NMR experiments in the Methods section on page 16 is misleading and does not reflect what we intended to convey. We have now rephrased the sentence on page 16 as shown below.

As explained on page 4 the addition of apyrase had little effect on mini-G_s binding measured by BLI, which is in agreement with the mutation in the mini-G proteins that made them insensitive to nucleotide binding when coupled to GPCRs (see ref 47). In agreement with this observation, none of our NMR measurements showed any GDP dependence. We have now added Supplementary Fig. 19 to demonstrate this for β_1 AR-E and β_1 AR-W.

For a better explanation of this we have therefore replaced the sentence on page 16 where it says “Variations in GDP binding to the G protein were mitigated through addition of 5 units of apyrase to degrade all GDP bound to the mini-G, and the sample was incubated for 2 hours at RT prior to measuring NMR spectra.” by “Spectra of ternary complexes formed using mini-G protein that was preincubated with apyrase (5 units, 2 hours) looked very similar to ones without apyrase treatment (Supplementary Fig. 19). Hence, all NMR spectra in this work were recorded without any prior treatment with apyrase, consistent with the BLI experiments.” We hope that this has clarified the issue.

4) lines 157-159: I fully agree with the authors assessment of M223 adopting a pre-active conformation in β_1 AR-W, compared to the active conformation in β_1 AR-E. It appears that the Iso-bound M223-W resonance almost sits on a linear trajectory between the Apo M223-E and the Iso-bound M223-E resonance, which would further support the assumption that M223-W is a pre-active state. It would be interesting to compare NMR spectra of partial agonist-bound β_1 AR-E to see where M223 would sit with respect to this trajectory.

We thank the reviewer for the interesting comment. As it turns out we are investigating this in a separate manuscript where we assess the effect of the activating mutation β_1 AR-E on partial agonists (in preparation).

In essence, as the reviewer rightly indicates, in the isoprenaline bound state the resonance M223-W sits indeed almost on a line connecting the resonances of M223-E in the apo form and M223-E bound to isoprenaline. This seems to further underpin the pre-active character of β_1 AR-W bound to isoprenaline. The partial agonist-bound M223-E resonances tend to be located along this line and loosely group between the apo-bound M223-E and the isoprenaline-bound M223-W signal, which is in agreement with the partial agonists populating the inactive/pre-active equilibrium. Interestingly, however, in contrast to M223-W where the M223 resonance positions were observed to correlate well with the efficacies of the partial agonists (Solt et al. Nat. Commun (2017) 8, 1795), for M223 in β_1 AR-E this correlation is less obvious. The signals are not as clearly on a line and the order is not always in agreement with the efficacy of the partial agonists used. This suggests that previously unseen contributions are now involved and that the activating mutation differentially affects the partial agonists from the full agonist. Although this is extremely interesting, we do not feel that this insight further contributes towards our manuscript. However, as mentioned already we have a manuscript on this topic in preparation.

5) lines 166-169: A recent publication (Chashmniam et al., 2021; ChemBioChem doi:10.1002/cbic.202000701) hypothesized that neighbour effects contribute more to the methyl ¹³C of Methionines than internal side chain dihedral angles. According to their theory, the upfield chemical shift change of iso-bound M223 in the 13C dimension may also be a result of the F^{6.44} shielding effect.

We thank the reviewer for their suggestion which we have now incorporated as a complementary interpretation on page 4. We have also included the reference to Chashmniam et al. in the manuscript.

The sentence on page 4 now reads, "Further, the large upfield ^{13}C shift of M296^{6,41} indicated a more gauche orientation of the χ^3 angle in the agonist-bound state, in agreement with the ternary complex structure but resulted possibly also from changes in shielding due to the changed orientation of F299^{6,44} (reference added citing Chashmniam et al.)".

6) line 180: I'm not sure if I agree with "reduced conformational variability". To me, the single (broadened) peak of M178 in the in the iso-bound $\beta_1\text{AR-E}$ spectrum suggests faster exchange rates compared to slow exchange in the iso-bound $\beta_1\text{AR-W}$ spectrum. What I find interesting, is that the minor M178 peak in the iso-bound $\beta_1\text{AR-W}$ spectrum (Supp Fig 4a) shifts to the $\beta_1\text{AR-E}$ position when bound to iso/Nb6B9 (Supp Fig 4b). Furthermore, the iso/Nb6B9 spectrum seems to show a minor M178 peak near the major peak observed in the iso-bound $\beta_1\text{AR-W}$ spectrum (Supp Fig 4a).

For the isoprenaline-bound spectra displayed in Supplementary Fig. 6a (previously 4a), we measure for the M178 signal linewidths at half-height for $\beta_1\text{AR-E}$ of (^1H 43 Hz; ^{13}C 50 Hz) and $\beta_1\text{AR-W}$ (^1H 46 Hz; ^{13}C 54 Hz). Whilst the two peaks are generally not too dissimilar, we note the slightly sharper signal for $\beta_1\text{AR-E}$ in contrast to the reviewer's comment of it being broadened. Accordingly, as we explain in the manuscript we associate the conversion of the two signals for M178 for $\beta_1\text{AR-W}$ into a single peak for $\beta_1\text{AR-E}$ with reduced conformational variability involving exchange that is slow on the chemical shift timescales and shifts the population towards a single conformation for M178 in $\beta_1\text{AR-E}$.

We agree with the reviewer that it is interesting to observe M178 of isoprenaline-bound $\beta_1\text{AR-E}$ to overlay well with the corresponding peak of Iso/Nb6B9-bound $\beta_1\text{AR-W}$. It suggests that the more activated receptor construct is able to accommodate the ligand in a form which to $\beta_1\text{AR-W}$ is only achievable once coupled to a cytoplasmic binding partner. We speculate that in $\beta_1\text{AR-W}$ complex formation takes place via the major form of M178, which then leads to a shift of the minor conformation of M178 towards the major form, until all is complexed. Of course, we can't exclude that this might happen directly as the reviewer seems to suggest.

The reviewer further points at the existence of a minor M178 peak in the iso/Nb6B9 spectrum that appears near the major peak in the iso-bound $\beta_1\text{AR-W}$ spectrum. We apologise that we omitted to label this signal (exact position ^1H 2.026 ppm/ ^{13}C 17.64 ppm) as artifact. We have now marked it with an asterisk in Supplementary Fig. 6b (previously 4b). The artifact is related to the intense M1/M283 signal and is the result of ^1H 180° pulse imperfections during the t_1 -MQ evolution period (see L.E.Kay J BNMR (2019) 73, 423-427). On our spectrometer this occurs sometimes due to power handling issues. In principle this can be eliminated through the use of a ^1H composite pulse during the t_1 -evolution period, however, long pulse lengths sometimes make the use of the latter difficult to accommodate during the initial t_1 period.

7) line 188-195: The authors claim that the more upfield ^1H chemical shift of M153 of iso-bound $\beta_1\text{AR-E}$ in Fig 2b is indicative of a more active conformation. However, the M153 ^1H chemical shift of Apo $\beta_1\text{AR-E}$ (Fig 2e) is located further upfield compared to iso-bound $\beta_1\text{AR-E}$ in Fig 2b. Does this mean that Apo $\beta_1\text{AR-E}$ is more active than iso-bound $\beta_1\text{AR-E}$?

Thank you for this comment, however, there seems to be some confusion here and we are unable to follow the reviewers' argumentation. Our statement on the observation of an upfield shift of M153 correlating with increased activity relates to ^1H chemical shifts as shown in Figure 2b and seems correct. The ^1H shifts for Apo and Iso. M153 in are 2.124 ppm and 2.068 ppm respectively, thus our statement stands. Figure 2e however shows ^{19}F chemical shifts of residue A282C. The latter have entirely different chemical shift dependencies.

8) line 194: I suggest labelling Fig 2d clearly as corresponding to the β_1 AR-E L289M mutant. A label in the figure will make it easier for the reader to rationalize the absence of the L289M resonance in Figs 2b & 2c.

We thank the reviewer for this suggestion. We have now clearly labelled each of the Figures 2b-d with the receptor construct under investigation i.e. Fig 2b-c are β_1 AR-E while Fig 2d is β_1 AR-E-L289M.

9) lines 198-200: It may be helpful to mention that agonist-bound X-ray structures of GPCRs often show TM6 in an inactive (or inactive-like conformation). Interestingly, the b36-m23 mutant used for crystallography (Ref 43) did not seem to harbor the E130W mutation, but Warne et al. argued that their thermostabilizing mutations resulted in preferential adoption of the inactive state regardless of the ligand pharmacology.

In response to the reviewers' comment we have now added the following sentence on page 5: "The latter is in agreement with X-ray structures of full agonist-bound receptors which frequently show TM6 in an inactive (or inactive-like) conformation." We have included a reference to GPCRdb.

The reviewer rightly mentions that the β_36 -m23 construct of β_1 AR which was used to solve the X-ray structure of the isoprenaline-bound receptor did not contain the W130 mutation. Nevertheless, the construct included six thermostabilising mutations (R68S, M90V, Y227A, A282L, F327A, F338M) that resulted from an alanine-scan which stabilized the receptor towards the inactive state. Accordingly, it is not entirely surprising that the cytoplasmic side of the receptor adopted the inactive state, even when bound to isoprenaline. We fully agree with the statement made by the authors of the X-ray study. The study is already cited in our manuscript (Warne et al. Nature (2011) 469, 241-245).

10) line 209: I understand that PDB 2Y03 is a "poor choice" for visualizing the active state as TM6 largely remains in an inactive conformation. It may be helpful to add an active (in complex with G_s) structure to visualize the extent of TM6 twisting/solvent exposure upon activation.

In response to this comment, we have now added a ternary complex structure into Supplementary Figure 7d (previously 5d). We note that the A282^{6,27} position is not resolved in the β_1 AR ternary complex structure with G_s (PDB: 7JJ0), so we have substituted this structure for the β_2 AR ternary complex X-ray crystal structure with G_s (PDB: 3SN6) in which this position is resolved.

11) lines 211-214: Why did the authors choose to do CPMG and not CEST experiments if ms-s timescale exchange is consistent with large TM6 helical movements?

We agree with the reviewer that CEST experiments would be applicable here, and we have now conducted experiments to examine if saturation transfer occurs for exchange between the active and pre-active states for β_1 AR-E A282C^{BTF_A,6.27} bound to isoprenaline at 298 K. Experiments were performed through the saturation of the active state peak shoulder and observing the changes to pre-active state peak intensity, relative to reference experiments saturating at an equal chemical shift difference on the other side of the pre-active state peak. The converse experiment (saturating pre-active and observing active) was also performed. Both experiments demonstrated reductions in peak intensities which were consistent with saturation transfer between the active and pre-active states (Supplementary Fig. 8), therefore indicating these states were in exchange on the ms-s timescale. See the Supplementary Fig. 8 legend and the methods for the experimental protocols. We have now changed these lines as follows: "Relaxation dispersion (CPMG) experiments did not show exchange on the μ s-to-ms timescale between the two states (Supplementary Fig. 7e). However, saturation transfer (CEST) ¹⁹F NMR experiments

demonstrated exchange between the active and pre-active states (Supplementary Fig. 8) consistent with slow ms-to-s exchange and helical movements, agreeing with L289M^{6,34} observations.”

12) line 215: Could the authors comment on why BTFA was used for TM6 and TET was used for TM7 labelling and/or add a reference to previous testing?

On page 15 in the Methods section (¹⁹F labelling) we already explain that we used BTFA labelling of A282C^{6,27} due to the "increased stability of the C-S bond formed therefore reducing problems due to rapid hydrolysis at the highly solvent exposed TM6 cytoplasmic tip." At 308 K the lifetime of the TET-label attached to A282C on TM6 was too short for our study of β_1 AR-E to be useful. We would have preferred to use TET due to the wider chemical shift dispersion. In contrast, hydrolysis is much less of a problem for the label at C344 so that for TM7 we were able to use TET as done previously in our work with β_1 AR-W (Frei et al. Nat. Commun. (2020) 11, 669). We now added, that for β_1 AR-E the position ^{TET}C344 on TM7 was "much less prone to hydrolysis".

13) lines 257-259: Do the authors have a means to estimate experimental errors of peak intensities shown in Supp Fig 6a? For example, if one condition was measured in duplicate, an average error can be determined and extrapolated.

We thank the reviewer for their suggestion. We have now added error bars representative of SD to Supplementary Fig. 9a (previously 6a).

14) line 265: Reference 59 refers to β_2 AR. Has this also been shown for β_1 AR?

We thank the reviewer for this question. Yes, this has also been shown for β_1 AR. For example, Warne et al. Science (2019) 364, 775-778. We have now added the reference in the text on page 6.

15) lines 306-308: How confident are the authors with the deconvolution? A residual error analysis (as described by Kim et al., 2013; JACS doi:10.1021/ja404305k) may be helpful.

We recognise the reviewer's concern that it is important to determine if the spectrum shown in Supplementary Fig. 9f (previously 6f) represents multiple peaks or a single peak of β_1 AR-E ^{TET}C344^{7,54}. In view of this, we are happy to provide further evidence that we believe supports our interpretation. We have now included single component deconvolutions for this data in Supplementary Fig. 10 with deconvolutions fit to minimise the residual on either the left- or right-hand side of the peak, with the centre of the simulated peak matching the centre of the raw data. This figure clearly indicates substantial residuals, which we believe indicates an asymmetric signal and therefore the data cannot be fit with a single signal. Additionally, we recorded a repeat of the same conditions (β_1 AR-E with isoprenaline and mini-G_s) on a Bruker Avance III 950 spectrometer (¹H 950 MHz) (The Francis Crick Institute, London) equipped with a 5 mm TCI HCN/z cryoprobe. The deconvolution of this signal into one or two major components (see below for Supporting Figure 1) also clearly indicates a single component is not sufficient to account for this signal as is supported by the residual larger than the noise. In contrast, with the residual smaller than the noise, the two component deconvolution supports two signals for the mini-G_s ternary complex. This agrees also with the data recorded at 600 MHz.

Supporting Figure 1. Deconvolution of the ^{19}F NMR spectrum for $\beta_1\text{AR-E}$ in the presence of isoprenaline and mini- G_s . The labelled receptor construct was $\beta_1\text{AR-E}^{\text{TET}}\text{C344}^{7.54}$. Raw data is shown in black. Individual simulated peaks are shown in green, and the sum of these peaks is shown in red. The residual (experimental spectrum minus sum of the simulated peaks) is shown in blue. Left, two ternary complex peaks were used to deconvolute the data. Middle, a single ternary complex peak was used to deconvolute the spectrum, with the simulated peak fit to minimise the residuals on the right-hand side of the peak and the centre of the simulated peak fit to the centre of the raw data. Right, the simulated peak fit minimised the residuals on the left-hand side of the peak. * indicates degradation products. The residual trace of the deconvolution assuming one signal is clearly larger than the noise level of the spectrum.

16) line 532: I would be more specific and use the term GPCRs instead of receptors.

Thank you, we have changed this now on page 12 in line with the reviewers' recommendation.

17) lines 555-557: The authors could employ a ^{19}F saturation transfer experiment as previously carried out by Frei et al., 2022; Nat Commun, doi: 10.1038/s41467-020-14526-3) to probe if a low-populated active state is present in the partial agonist-bound $\beta_1\text{AR-E}$ solution ensemble.

We agree with the reviewer that a CEST experiment here would be useful to probe for low-populated active state. Following a similar strategy to the experiment in Supplementary Fig. 8, we performed CEST experiments on $\beta_1\text{AR-E A282C}^{\text{BTFA},6.27}$ with partial agonist xamoterol at 298K, saturating at the anticipated active state position from the spectra with full agonist isoprenaline. This experiment showed negligible saturation transfer, suggesting the active state was not (or insufficiently) populated (Supplementary Fig. 13a). Although there are possibilities that the exchange is not observed because the active state is too low in population or exchanges with the pre-active state significantly slower than for isoprenaline-bound receptor, the simplest interpretation of this experiment is that the active state is not populated. Accordingly, for the ternary complex of $\beta_1\text{AR-E A282C}^{\text{BTFA},6.27}$ with partial agonist xamoterol and mini- G_s (0.75 molar equivalents), we conducted a comparable experiment where we saturated the pre-active state to establish if ternary complex formation resulted from the pre-active state. We conducted this experiment at 308 K to increase the separation of the signals. We observed chemical exchange between the partial agonist bound pre-active state and the ternary complex (Supplementary Fig. 13b). In view of the established absence of the active state, this suggests that complex formation indeed occurs via the pre-active state as proposed in our manuscript. We have now added: " ^{19}F NMR saturation transfer experiments with $\text{A282C}^{\text{BTFA},6.27}$ confirmed that the ternary complex TM6 signal exchanges with the pre-active state signal (Supplementary Fig. 13b)."

18) line 572: A space is missing after "conformation".

Thank you, this has been corrected.

19) lines 572-575: I suggest a more granular discussion of the authors findings in comparison with reference 27. The cryo-EM structures in reference 27 used heterotrimeric G proteins while the authors used mini-G α subunits. The simplified system using only the mini-G α subunits may very well indicate similar koff rates for canonical and non-canonical G proteins, but the cryo-EM structures in reference 27 clearly suggest G protein subtype-specific β_1 AR:G protein interaction interfaces that extend beyond the α 5 helix. Hence, off rates for heterotrimeric G proteins (or wild type G α subunits) may differ from those observed for the mini-G's.

It may be useful to compare the effect of the mini-G proteins used in this study with that of the wild-type heterotrimeric G proteins on ligand binding (like Han et al., 2023 (Ref 35); Extended Figure 10c). Such an experiment could inform on how well the engineered mini-G proteins reflect their wild type counterparts and potentially give the authors more confidence in their claim that G protein on rates are the main drivers of G protein subtype selectivity even if the full heterotrimers were used. This could also be done in the presence and absence of nucleotides (here I'm referring to lines 601 – 607).

Our data indicates that the G α subunits contribute similarly to the overall stability of the final complex, indicating that the differences in the α 5 helix provide similar contributions to the final complex stability despite the differences in sequence. The mini-G proteins represent a majority of the GTPase domain of the G α subunit. The primary deletion is the alpha helical domain, which does not make contacts to β_1 AR in cryo-EM structures (PDB: 7JJO for G $_s$, 7S0F for G $_i$). Therefore, wild type G α subunits are likely to show similar affinities. The reviewer quite rightly suggests that this assumption requires investigation, but we note that the mini-G proteins replicate a majority of the contacts with receptors (97% according to Carpenter, B., & Tate, C. G. (2016). *Protein Engineering, Design & Selection*, 29, 583–593.). Interestingly, there are differences between the G α subunit interfaces with β_1 AR between G $_s$ and G $_i$ (1,104 Å² and 923 Å², respectively, Alegre, K. O., et al. (2021). *Nature Structural & Molecular Biology*, 28, 936–944), and therefore we suggest that our data using mini-G proteins, as good mimetics of this interface, implies these differences in surface area don't affect complex stability – i.e. the differences in interactions compensate for the differences in the interface size. We also acknowledge that there are differential contributions of the G α and G β subunits to the interface with receptor between G $_s$ and G $_i$ as found in Alegre et al., 2021. This too could influence the complex stabilities and should also be investigated.

Ref. 35 (Han et al., 2023) does not provide more than the G protein binding affinity (K_B), and thus investigation of the k_{on} and k_{off} has not been established for the κ -opioid receptor. There are also likely differences between the affinities for G proteins between receptors, so a direct comparison is unlikely to be useful. Despite this, the G protein binding affinities measured were similar to the orders of magnitude of K_D values obtained from our measurements (341 nM to 1,551 nM), and therefore potentially hints at a similar pattern between receptors and heterotrimeric G proteins.

The investigation of wild type heterotrimeric G protein affinities for receptors is beyond the scope of this study, however we agree that a more detailed study into the varying affinities between receptors and G proteins is important to establishing the mechanisms of G protein selectivity. We are actively working on follow-up studies with heterotrimeric G proteins to examine the effects of the $\beta\gamma$ subunit and the effects of GTP on complex dissociation. Our BLI methodology offers a direct measure of ternary complex affinities, and therefore would be preferable for measuring the kinetics of heterotrimeric G proteins to ligand binding studies as in Ref 35. The authors of that study also recognise the importance of the measurement of k_{on} and k_{off} parameters, not just K_D , in their discussion.

With the reviewer's comment in mind, we have now changed this section of the discussion to:
"Cryo-EM structures of β_1 AR reveal a larger interface size with G α_s than G α_i , but our dissociation kinetics data suggested complex stability was comparable between all ternary complexes with mini-G proteins. This indicated the overall conformation and differing interacting surface and residues between the mini-G proteins provided similar overall enthalpic contributions to the final ternary complex."

We have removed the reference to Ref 35, as this study examined the effects of GDP and GTP on ligand binding, and thus did not directly measure the effects of ternary complex dissociation rates. We thought referencing this work in relation to dissociation rates would be misleading.

Additionally, we have added a phrase into the line on page 13 : “Whilst our study was conducted with structurally validated mini-G proteins and fully functional receptor with minimal thermostabilisation, additional kinetic studies using full length receptor inclusive of full-length IL3 and C-terminus with heterotrimeric G proteins, including the $\beta\gamma$ -subunits which subtly and differentially contribute to interaction interfaces, could aid in revealing further G protein family dependent kinetic differences.”

20) lines 583-586: I suggest using “is in partial agreement” since reference 69 also highlights the importance of other Ga subunit regions in G protein subtype specificity.

We have now incorporated the suggested change on page 13.

21) lines 586: Insert “is” between selectivity and in.

The change has been made.

22) line 591: I suggest writing “was shown to significantly reduce” instead of “was significantly shown to reduce”.

We have included the suggested change on page 13.

23) line 622: I suggest to either delete “mildly” or change to “minimally”.

On page 14 we now say ‘minimally’ instead of ‘mildly’.

24) lines 728-729: The authors state that 2 molar equivalents of mini-G were used. The Figure legend of Figure 3c, however, states that 8 equivalents of mini-Gs were used in combination with xamoterol and the Figure legend of Figure 5d states that 8 equivalents mini-Gi1 were used. Furthermore, I would recommend using “molar equivalents” instead of just “equivalents” in the Figure legends.

In the Methods section (NMR experiments) on page 16 we clearly state that we used 2 or 8 molar equivalents, which is in agreement with the descriptions provided in the legends to Figure 3 and 5. The exact values used depend on the affinity of the complex and are provided in the legends to the figures.

We have now changed "equivalents" to "molar equivalents".

25) lines 806-807: The authors mention “varying binding partner concentrations”. However, Extended Figure 2E only shows representative data using 2 μ M mini-G. Were all BLI experiments carried out at a single mini-G concentration of 2 μ M?

In the Methods section lines 806-807 the 'varying binding partner concentrations' statement refers to the fact that different mini-G binding partners required different concentrations to obtain a BLI binding trace (in view of the 1-2 order of magnitude difference in K_D values between the different mini-G complexes, different ligands etc). The statement is not suggesting that dose response curves were recorded. We have

now clarified this in the light of our response to reviewer #2 and also to your comment point 2 (see above). We provide additional Supplementary Fig. 17 showing all binding curves and we provide a clear description of the experimental conditions in Supplementary Table 4, which includes the mini-G concentrations used to obtain the kinetics parameters. We have also added a Supplementary Methods section to clarify our proceedings.

With regard to the reviewers' comment, the 2 μ M of mini-G_s mentioned in the legend to Supplementary Fig. 2e refer to an experiment where we compare ternary complex formation between apo and isoprenaline-bound receptor.

26) Figure 1d: Please state that the BLI binding traces show mini-Gs binding to the b1AR constructs.

We have now labelled Figure 1d with mini-G_s (similar to how it is presented in Figure 3b) and we mention the use of mini-G_s (inclusive of concentrations) also in the legend to the figure.

27) Figure 3: The legend states that for c) 8 equivalents of mini-Gs were used in combination with xamoterol while in f-h) 2 equivalents were used in combination with all ligands including xamoterol. Could the authors explain why different molar equivalents of G protein were used in combination with the same ligand?

This is an error in the text. The spectra shown in Figures 3f-h were recorded with identical molar equivalents of mini-G_s as for the spectra shown in Figure 3c. Accordingly, 2 molar equivalents of mini-G_s for the complex with isoprenaline, 8 molar equivalents of mini-G_s for the partial agonist salbutamol and xamoterol, respectively. We have now corrected this in the Figure legend to Figure 3f-h.

28) Figures 3b and 4: What do the error bars describe, SD or SEM? How many biological (n) and technical replicates were collected?

In Figure 3b the error bars indicate SD (3 technical replicates).

In Figure 4a and 4b the error bars indicate SEM (3 technical replicates).

In Figure 4c the error bars indicate SD (3 technical replicates).

29) Supplementary Figure 2F: KD value incorrect

The K_D value had a typo. However, in the light of providing a BLI trace of improved quality i.e. that clearly approaches zero intensity at extended times, we repeated the measurement in the presence of apyrase and provide an updated Supplementary Fig. 2f. We have updated the values for K_D , k_{on} and k_{off} in Supplementary Fig. 2f. As before the data show that there is virtually no dependence on the presence of apyrase.

30) Supplementary Figure 9: Are those biological (n) or technical triplicates?

These are technical triplicates.

REVIEWER COMMENTS

Reviewer #1 (Remarks to the Author):

The authors have responded in detail to all my comments in a very satisfactory manner. The issues raised from my side are thus all addressed.

Reviewer #2 (Remarks to the Author):

Dear corresponding author

I find your chosen strategy to be robust and supported by sufficient evidence and data. The comparison of the 'optimised' single-point measurement technique against the gold standard of full dose-response analysis demonstrates consistency in obtaining reliable binding parameters. Your approach of individually pre-inspecting concentration series for mini-G proteins, ensuring a 1:1 monophasic behavior, is sound and effectively addresses challenges posed by weaker-binding proteins. This method apparently allows for a comprehensive comparison of mini-G complex formation across various affinities while maintaining protocol consistency. I also concur that there is only negligible impact on your main findings due to substantial numerical differences in k_{on} and KD values. Overall, the 'optimised' single-point method appears to be effective, yielding highly reproducible results.

Thank you for your prompt response and clarification regarding the BLI protocols. The inclusion of a scheme outlining the assay procedure is appreciated and has indeed provided valuable insights into how non-specific interactions are mitigated. The pre-loading steps, with the incubation in buffer containing 0.1% BSA to block non-specific interactions, and the subsequent mock association step with agonist for further blocking, contribute to the robustness of your approach. The demonstration of minimal weakly bound material and the reference channels confirming the absence of non-specific interactions are valuable additions.

Thank you for addressing the ambiguity surrounding the reference channel. Your clarification regarding its role, including identical conditions to experimental channels, confirms the specificity of receptor loading as biotin-specific. The use of the reference channel to assess non-specific association, as depicted in Supplementary Fig. 3e, and subsequent subtraction of this interaction from experimental channels, adds clarity to your methodology.

I'm very pleased to see that you reached out to obtain additional reagents for testing in the assay to demonstrate specificity. The control experiment using the avi-tagged and biotinylated construct of the SARS-CoV-2 Spike RBD protein is indeed a valuable addition. The obtained data further supports the specificity of the interaction, especially when considered alongside minimal interaction in the absence of agonist.

Overall, I am pleased to acknowledge that all the concerns and suggestions raised in my review have been thoroughly and comprehensively addressed. You have taken proactive measures, either through the execution of additional experiments or by providing detailed scientific explanations, to enhance the clarity and robustness of the manuscript. Based on your responses and the modifications made, I am confident in expressing my satisfaction with the current form of the manuscript. I believe it is ready for publication as is, given the thoughtful attention to the feedback provided, that also includes the responses to the feedback from my unknown co-reviewers.

Reviewer #3 (Remarks to the Author):

I am very satisfied with the authors' responses concerning NMR experiments.

However, I still have a few questions regarding the pharmacological data.

A. In cell ligand affinity measurements appear to be collected as biological triplicates since the methods section states "...represented as mean of three separate experiments \pm SEM" and the legend of Figure 1 states "Values are averages of three separate measurements and standard deviations are shown." This raises the following questions: a) are the error bars in Figure 1a SEM (as stated in methods) or SD (as stated in the figure legend)? and b) were the individual measurements done as technical replicates?

B. In the legend of Figure 4 of the revised manuscript, it is now indicated that TRUPATH assays (Fig 4a, b) are presented as technical (i.e. not as biological) repeats. In the TRUPATH assay methods section, however, it is indicated that TRUPATH data is "represented as mean of three experiments \pm SEM." This raises the following questions: a) are the data shown in Figures 1b, 4a, 4b the mean of biological (as stated in methods) or technical replicates (as stated in Figure legends)?, b) are the error bars in Figure 1b SEM (as stated in methods) or SD (as stated in the figure legends)? and c) if the data represents biological replicates, were the individual measurements done as technical replicates?

It would be surprising if the In cell ligand affinity and TRUPATH experiments were only done as technical repeats, as it is common practice to conduct pharmacological experiments as biological repeats (each in technical duplicate or triplicate). Can the authors please clarify?

C. In legends of Figures 3b & 4c of the revised manuscript, it is now indicated that BLI experiments were collected as technical but not as biological repeats. Performing biological repeats of the BLI experiments (particularly since only single concentrations are measured) would provide a stronger statistical basis. Ideally, these would be done using separate receptor and G protein preparations.

Point-by-point reply to the reviewers' comments: Second Revision

We would like to thank the reviewers for their replies to our response, and for their strong endorsement of our paper. We are grateful for their thorough and detailed responses, and we believe that the changes we have made in response to their comments have considerably improved the quality of our submitted work.

We are pleased that reviewers #1 and #2 are satisfied with the changes we have made. We are thankful to reviewer #3 for their reply to our responses and appreciate their further suggestions. We hope to provide clarity on the additional points raised.

Reviewer #3

A) a) Are the error bars in Figure 1a SEM (as stated in methods) or SD (as stated in the figure legend)? and b) were the individual measurements done as technical replicates?

and

B) a) Are the data shown in Figures 1b, 4a, 4b the mean of biological (as stated in methods) or technical replicates (as stated in Figure legends)?, b) are the error bars in Figure 1b SEM (as stated in methods) or SD (as stated in the figure legends)? and c) if the data represents biological replicates, were the individual measurements done as technical replicates?

We apologise for the confusion and thank the reviewer for highlighting the discrepancy between the methods and the figure legends. The reviewer is quite correct that all the pharmacology data (*in cell* ligand affinity measurements and TRUPATH assays) were the mean \pm SEM of $n=3$ independent biological experiments each performed in duplicate as described in the methods. The legends have therefore been amended according. Furthermore, whilst the reviewer hasn't commented on the supplementary figures, we have also revised the Supplementary Information to include this for Supplementary Figure 14.

C. In legends of Figures 3b & 4c of the revised manuscript, it is now indicated that BLI experiments were collected as technical but not as biological repeats. Performing biological repeats of the BLI experiments (particularly since only single concentrations are measured) would provide a stronger statistical basis. Ideally, these would be done using separate receptor and G protein preparations.

We thank the reviewer for their comment on the BLI data. Our experiments were performed in triplicate, in that three individual tips were used to generate the experimental channels, each dipping into separate wells for receptor immobilisation and G protein binding affinity measurements. An example of the layout and protocol is shown in Supplementary Fig. 3a. Each of the three biosensors generated individual isotherms, each of which were then used to calculate binding kinetics. Therefore, each of the measurements in the triplicate were generated using different immobilised receptor populations (i.e. different biosensor tips) and different mini-G protein populations. Means and SDs of the $n=3$ individual repeats were then calculated for each kinetic parameter. We have modified the Supplementary Methods section to hopefully clarify the protocols we used in collecting these data. We have modified the legends in Figures 1, 3 and 4, as well as in relevant Supplementary Figures, which now read as "three individual repeats".

The components in the wells were prepared from the same concentrated stocks of purified proteins between the three replicates. We agree with the reviewer that "Performing biological repeats of the BLI experiments" would be "stronger statistical basis". However, due to availability of resources we have only

performed the n=3 individual repeats as described above. We believe we have provided quality control and validation of the functionality and integrity of the proteins used in this study, and as such that these n=3 individual repeats are sufficient to support our findings.

We hope that the edits we have added, shown in green, now add clarity to the manuscript and address all the reviewers' comments.